# A Scene is Worth a Thousand Features: Feed-Forward Camera Localization from a Collection of Image Features

**Axel Barroso-Laguna**[1]   **Tommaso Cavallari**[1]   **Victor Adrian Prisacariu**[1,2]   **Eric Brachmann**[1]
[1]Niantic Spatial    [2]University of Oxford          https://nianticspatial.github.io/fastforward/

## Abstract

Visually localizing an image, *i.e.*, estimating its camera pose, requires building a scene representation that serves as a visual map. The representation we choose has direct consequences towards the practicability of our system. Even when starting from mapping images with known camera poses, state-of-the-art approaches still require hours of mapping time in the worst case, and several minutes in the best. This work raises the question whether we can achieve competitive accuracy much faster. We introduce FastForward, a method that creates a map representation and relocalizes a query image on-the-fly in a single feed-forward pass. At the core, we represent multiple mapping images as a collection of features anchored in 3D space. FastForward utilizes these mapping features to predict image-to-scene correspondences for the query image, enabling the estimation of its camera pose. We couple FastForward with image retrieval and achieve state-of-the-art accuracy when compared to other approaches with minimal map preparation time. Furthermore, FastForward demonstrates robust generalization to unseen domains, including challenging large-scale outdoor environments.

## 1 Introduction

Humans understand complex 3D scenes in seconds. With a glance at a few images of any environment, we can form a mental map and reckon where each image was taken. This inherent ability to localize in a scene allows us to navigate and understand our surroundings with ease. However, replicating this intuitive process within an algorithm, *i.e.*, a visual localizer, is challenging. Visual localizers provide camera location and orientation enabling real-time applications like navigation or Augmented Reality (AR), but they require more than just a few seconds of looking at the scene to be able to do it (Brachmann et al., 2023).

One popular family of visual localization approaches requires knowing the structure of the scene, and therefore, before being able to locate an image in the environment, they build a 3D model of the scene (Humenberger et al., 2020; Sarlin et al., 2019; Sattler et al., 2016). Such structure-based localizers find correspondences between 3D scene points and 2D query image points and solve for the pose using algorithms like PnP-RANSAC (Gao et al., 2003; Fischler & Bolles, 1981). These methods rely on structure-from-motion (SfM) pipelines to build the 3D representation of the scene and the runtime of every scene depends highly on the number of images, ranging from minutes to hours for a few hundred mapping images (Schonberger & Frahm, 2016).

To address these limitations, scene coordinate regression (SCR) (Li et al., 2020; Brachmann et al., 2017; 2023) and absolute pose regression (APR) (Kendall & Cipolla, 2017; Shavit et al., 2021; Chen et al., 2024; 2022) methods optimize a neural network to learn an implicit representation of the scene, inferring dense scene coordinates (SCR) or absolute poses (APR) from unseen query images. The mapping time corresponds to the network training time, which has been reduced to minutes in recent approaches (Brachmann et al., 2023; Chen et al., 2024). They offer accuracy comparable to structure-based localizers but require dense training coverage and generalize poorly to unseen areas. Alternatively, relative pose regression approaches (RPR) estimate poses between query and retrieved images without per-scene training or 3D map preparation (Balntas et al., 2018; Arnold et al., 2022; Zhou et al., 2020). These approaches are attractive since they significantly

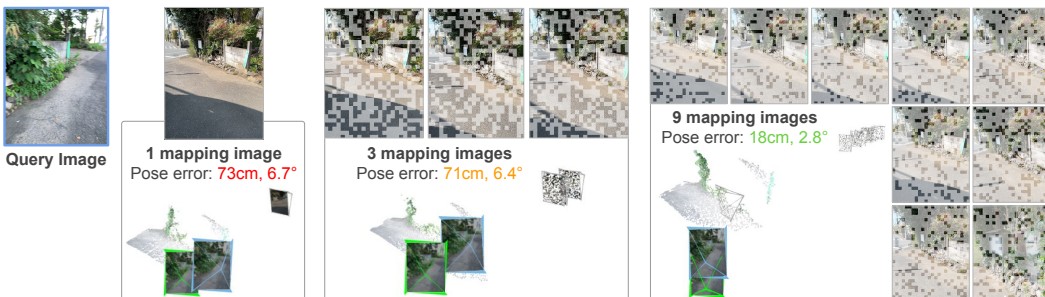

Figure 1: We introduce **FastForward**, a network that predicts query coordinates in a 3D scene space relative to a collection of mapping images with known poses. FastForward represents the scene as a random set of features sampled from mapping images, and returns the estimate for a query w.r.t. all mapping images in a single feed-forward pass. From left to right, we show how results improve when FastForward uses an increasing number of mapping images, as returned by image retrieval. Note that we always sample the same number of mapping features, and hence, FastForward's query runtime and GPU memory demand remains roughly constant in all three examples.

reduce the mapping requirements by relying only on images and poses, which are obtainable via real-time systems, *e.g.*, SLAM (Murai et al., 2025). However, RPR methods generally lack the accuracy of structure-based or SCR competitors. Other works propose to mitigate the RPR's low accuracy by triangulating local point clouds from retrieved images (Sattler et al., 2017; Torii et al., 2019), however, their post-processing steps result in significantly longer localization run-times than standard RPR methods.

In this work, driven by the motivation of reducing the overhead of mapping to a minimum, we propose FastForward, a novel approach that achieves fast mapping and localization through a single feed-forward pass. FastForward takes inspiration from recent foundation models (Kirillov et al., 2023; Wang et al., 2024b) and scene representation networks (Jin et al., 2024a; Sitzmann et al., 2021), which have shown strong performance and outstanding generalization capabilities across tasks and datasets, and have pushed the boundaries of what we thought was possible just a few years ago (Leroy et al., 2024; Wu et al., 2024). This success motivates our next question: What is the minimal map representation that enables accurate and efficient visual localization? We claim that a collection of image features encoding local visual appearance as well as their 3D locations within the scene is a powerful and convenient map representation. Our architecture design is inspired by DUSt3R (Wang et al., 2024b), but instead of taking two images as input, FastForward takes one query image as input as well as a random sample of features from multiple posed mapping images. Thereby, we predict accurate query 3D coordinates directly in the map coordinate system, see Figure 1. Since FastForward has access to a *collection* of features spanning multiple mapping images, it avoids the need for computing relative pose estimates between the query and multiple mapping images, one by one. Different from binocular RPR methods that rely on heuristics for scale-metric pose estimates (Arnold et al., 2022), FastForward transfers the correct scale directly from the mapping poses, even enabling it to generalize to scene scales not seen during training.

We summarize our **contributions** as follows: **1)** We demonstrate that a scene representation consisting of only a few hundreds mapping features is sufficient for fast and accurate visual localization. **2)** We present FastForward, a simple yet effective architecture that enables localization of an image relative to a collection of mapping features in a single feed-forward pass. **3)** A scene and scale normalization approach within the architecture that boosts the generalization capability in domains with different scale ranges for image localization.

## 2 RELATED WORK

Visual localization methods require knowing the structure of the environment, and hence, before being able to locate a new query image in the scene, they need to define how they represent the scene in which they want to localize.

**Structure-based Localization** requires building a 3D model of the scene. These models are typically created by SfM software (Humenberger et al., 2020; Schonberger & Frahm, 2016; Pan et al., 2024). At localization time, these approaches first establish correspondences between the query image and the pre-built 3D model by keypoint matching (Lowe, 1999; Barroso-Laguna et al., 2019; 2020; 2022; Tian et al., 2020b;a; DeTone et al., 2018), and then, solving for the query pose with a robust estimator (Barath et al., 2019; 2020; Barath & Matas, 2021; Chum & Matas, 2005; Barroso-Laguna et al., 2023). While these methods can be efficient at inference time (Lindenberger et al., 2023; Wang et al., 2024c), feature triangulation with SfM can take several hours, or even days, depending on the number of mapping images.

**Scene Coordinate Regression** methods regress the 3D coordinates in the scene space for the 2D pixels of a query image (Shotton et al., 2013). The output and the input to the SCR algorithm already establish the 2D-3D correspondences. A robust estimator can be applied as in the case of structure-based localization to compute the query pose. Traditionally, SCR relied on random forest (Shotton et al., 2013; Valentin et al., 2015; Brachmann et al., 2016; Cavallari et al., 2017; Cavallari et al., 2019), but in recent years, SCR improved their accuracy by employing convolutional neural networks (Brachmann & Rother, 2021; Cavallari et al., 2019; Li et al., 2020; Dong et al., 2022). The map representation of the scene is implicit, and in the case of a neural network, is encoded in its weights. One traditional limitation of SCR is the time to train such networks. Recently, ACE (Brachmann et al., 2023) proposed a patch-based training scheme that addressed that issue reducing the training time to 5 minutes. GLACE (Wang et al., 2024a) improves the accuracy of ACE in large areas, but it also increases its training time to 25 minutes. NeuMap (Tang et al., 2023) encodes a scene into a set of map codes and uses a coordinate regressor to estimate the query scene coordinates. Their regressor network is trained per dataset, and map codes trained per scene, taking considerable time to optimize. Furthermore, NeuMap requires a pre-built 3D model to initialize their system. Different from SCR methods, FastForward is pre-trained on a large-scale dataset, and requires no further scene-specific training.

**Relative Pose Regression** systems aim at localizing a query image by regressing the relative pose between the query and the most similar (or top-K) mapping images (Ding et al., 2019; Zhou et al., 2020; Winkelbauer et al., 2021; Arnold et al., 2022). Adding more mapping images enables more precise absolute positioning through multi-view triangulation (Laskar et al., 2017; Zhou et al., 2020; Winkelbauer et al., 2021). An attractive characteristic of RPR methods is that they do not require any scene-specific training. Our approach shares a core principle with RPR methods: it estimates the query pose relative to a map representation. However, while RPR methods rely on a single reference image, we represent the map as a collection of 3D-anchored mapping features.

**Semi-generalized Relative Pose Estimation** methods compute the absolute pose of a query camera relative to a generalized camera composed of multiple mapping images with known poses (Bhayani et al., 2021; Panek et al., 2024; 2025). This formulation allows for the recovery of the absolute translation scale from additional mapping images. An efficient implementation of this approach is the E5+1 solver, which utilizes five point correspondences between the query and one mapping image to estimate the essential matrix, and a single additional correspondence with a second mapping image to resolve the scale (Zheng & Wu, 2015). Such methods typically rely on RANSAC-wrapped geometric solvers operating on 2D-2D image pair matches rather than exploiting the multi-view structure and relationships in a single feed-forward pass, leading to lower pose accuracy than FastForward in unstructured or challenging scenarios.

**Foundation Models.** Large neural networks have seen an enormous advancement thanks to the scalability of new architectures. These models, based on transformer networks (Vaswani et al., 2017), are trained on large-scale datasets, and have proven to have very strong generalization capabilities as well as outstanding performance. One example of these foundation models is DUSt3R (Wang et al., 2024b), which takes two images as input and addresses different two-view problems by simplifying them into a single task: the prediction of aligned point maps. The simplicity and accuracy of DUSt3R have motivated many follow-up works. MASt3R (Leroy et al., 2024) builds on top of DUSt3R by adding a descriptor head that improves correspondence accuracy through descriptor matching. MASt3R-SfM (Duisterhof et al., 2024) embeds MASt3R into an SfM pipeline, Stereo4D (Jin et al., 2024b) introduces an extension for dynamic scenes, and Wang & Agapito (2025); Yang et al. (2025); Elflein et al. (2025); Wang et al. (2025b;a) present modifications to the original DUSt3R to enable multi-view 3D reconstruction. Viewformer (Kulhánek et al., 2022) uses a transformer architecture that, given multiple posed images, creates a code representation that can

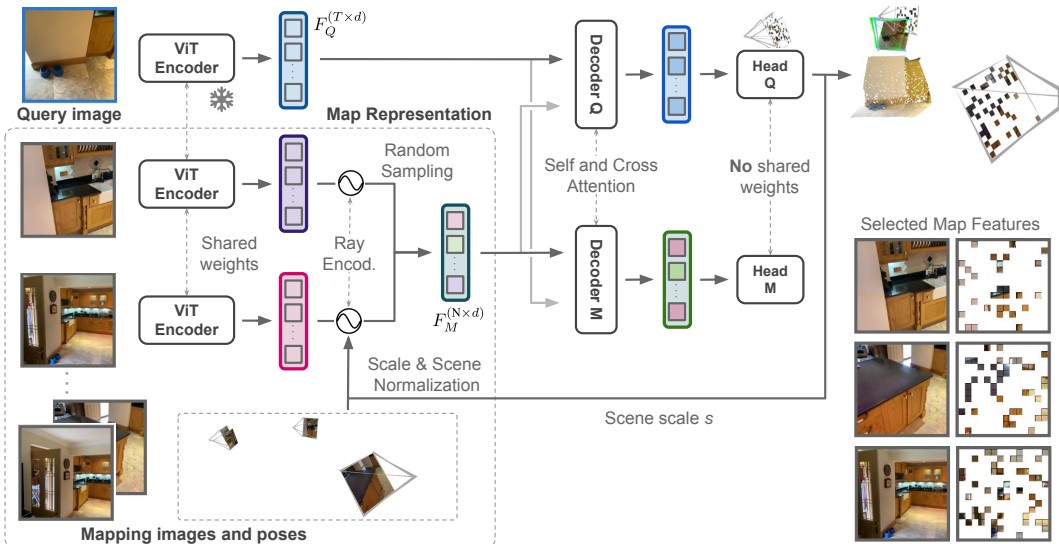

Figure 2: **FastForward Architecture**. FastForward uses a ViT encoder to compute features of the query, $I^Q$, and the mapping images. To create the map representation M, we randomly sample N mapping features. Each mapping feature is augmented with a ray embedding that encodes its camera's position and viewing direction. Mapping poses are normalized by setting one pose to the origin and defining the maximum translation in any direction to one. FastForward performs self- and cross-attention between the query features and the map representation. The query head predicts the 3D coordinates of the query features in the normalized space. The metric scale is recovered by applying the scene scale factor ($s$). The predicted 2D-3D correspondences yield the final query pose ($P_Q$). During training, a mapping head also predicts 3D coordinates for the mapping features, providing additional supervision.

be used either for novel view synthesis or image localization. Their work is primarily designed for the novel view synthesis task and lags behind current localization baselines. Reloc3r (Dong et al., 2024) recently demonstrated that a symmetric DUSt3R can significantly improve RPR's accuracy. However, Reloc3r still relies on two-view relative pose estimates. FastForward leverages a more robust multi-view scene representation, which allows it to outperform RPR methods and even achieve competitive or superior accuracy to SCR and structure-based algorithms in certain scenarios.

## 3 METHOD

Given a database of posed mapping images from a scene, $\mathrm{M} = \{I_k \in \mathbb{R}^{H \times W \times 3} \mid k = 1, ..., K\}$, our objective is to estimate the position and orientation of a new query image, $I^Q$, with respect to M. We define the camera pose, $P_Q$, as the rigid transformation that maps coordinates from the camera space to the scene space. First, we use the images in the database M to define the map representation, which is fed into a transformer network together with query features to predict query 3D coordinates, as seen in Figure 2. At inference time, we utilize the predicted 3D coordinates to define 2D-3D correspondences and compute the pose $P_Q$ through PnP-RANSAC (Gao et al., 2003; Fischler & Bolles, 1981).

### 3.1 MAP REPRESENTATION

We aim to minimize the computational and time requirements for localizing a query image. One shared characteristic of modern visual localization systems is extracting neural network features from the mapping images. These features are then utilized to train a specialized neural network (SCR), triangulate 3D points (SfM), or serve as input to a subsequent network that computes the relative pose (RPR). Such features are a powerful representation of the scene, but they are also heavy to process if we were to use all of them directly when localizing a new query image.

**Feature Sampling.** Transformer models look at the whole image before updating its features, and therefore, contrary to previous CNN-based feature extractors, transformers provide features with a global context. For extracting features, we adopt a ViT encoder (Dosovitskiy et al., 2020), which tokenizes the images and extracts features $F^k$ from $I^k$. The encoder produces rich features, but also some redundant features for visual localization because similar areas in the image might not provide new information. We show that just a few features from the images are enough to represent the scene M. This is advantageous both during mapping and inference. During mapping, only the image feature extraction step is required. At inference time, it scales well to a growing number of mapping images by fixing the size of the map representation N, *i.e.*, the total number of features we use from the mapping images. Since we do not know which regions of the mapping images are relevant for a new query, we randomly sample on the set of mapping features as seen in Figure 2.

**Scene and Scale Normalization.** Scale estimation is an ambiguous problem in 3D computer vision. When only having access to images, the information is limited to the 2D plane, and the true distance between the cameras remains unknown (Tateno et al., 2017; Arnold et al., 2022). Multi-image methods must distill the scale of the scene from the mapping poses to guarantee multi-view consistency. However, this is challenging when training across multiple datasets that display different scale ranges. To help the network generalize to new domains and exploit metric and non-metric training data, we adopt a simple yet effective scene normalization technique. We normalize the scene by defining one of the mapping images, $I^0$, as the reference, and transform all other mapping images such that $\bar{P}_k = P_0^{-1} P_k$. This places the scene at the origin of the coordinate system. The network is tasked to predict query coordinates in the normalized scene. Furthermore, as in Guizilini et al. (2025), we also normalize the scale of the mapping cameras. We compute the scene scale s as the largest camera translation in any of the spatial coordinates after scene normalization, *i.e.*, $s = \max\{|x|, |y|, |z|\}_k^K$, where $t = [x, y, z]^T$ is the translation component of the mapping pose $\bar{P}_k$. We normalize all camera translations such that $\hat{t} = [x/s, y/s, z/s]^T$. Once the network predicts the 3D coordinates, we multiply them by $s$ to recover the true scale of the scene. In this way, we abstract the task of learning metric coordinates from the poses and images. As seen in the Appendix C.1, scale normalization makes the network more robust to scale ranges not seen during training.

**Ray Encoding.** To inform the network about the origin of each mapping feature $f_{ij}^k$, we use a ray encoding that represents its 3D position and orientation in the normalized scene. Specifically, we use a Fourier encoding (Mildenhall et al., 2021) to tokenize the mapping cameras. Each camera is parameterized as a ray vector containing the origin $\hat{t}_k = [x, y, z]^T$, and its viewing direction $r_{ij}^k = (\mathbf{K}_k \mathbf{R}_k)^{-1}[u_{ij}, v_{ij}, 1]^T$, where $u_{ij}, v_{ij}$ represent the center pixel of the feature token $f_{ij}^k$, $\mathbf{K}_k$ are the camera intrinsics, and $\hat{t}_k$ and $\mathbf{R}_k$ are the translation and rotation component of the mapping image $I^k$. Finally, we use an MLP layer to project the encoding to the same dimension as the feature vector $f_{ij}^k$, obtaining the ray encoding $R_{ij}^k \in \mathbb{R}^{N \times d}$.

## 3.2 ARCHITECTURE

**Encoder-Decoder.** As discussed, we utilize the ViT (Dosovitskiy et al., 2020) architecture to tokenize the input images. We initialize the encoder with a pre-trained DUSt3R model (Wang et al., 2024b) and freeze its weights during training. We process the image tokens through multiple ViT blocks, composed of self-attention and MLP layers. An image $I \in \mathbb{R}^{H \times W \times 3}$ results in a feature map $F \in \mathbb{R}^{T \times d}$, where $T = \frac{H}{16} \times \frac{W}{16}$ and $d = 1024$. The map representation is generated by sampling N features from the collection of mapping features and fusing them with the ray encodings, such that $F_M = \{R_n + f_n \mid n = 1, ..., N\}$. For the decoder, we use ViT blocks initialized from DUSt3R and fine-tune them during training. The decoder incorporates cross-attention blocks between the self-attention and the MLP layers. The cross-attention allows the network to reason about the structure of the scene and its relationship with the query image. This reasoning occurs within a single forward pass, enabling the map representation to adapt based on the query image features. We obtain the final query and mapping features as:

$$\bar{F}_Q^{(T \times d)} = \text{Decoder}_Q(F_Q^{(T \times d)}, F_M^{(N \times d)}), \text{ and } \bar{F}_M^{(N \times d)} = \text{Decoder}_M(F_M^{(N \times d)}, F_Q^{(T \times d)}). \quad (1)$$

**Heads.** We follow recent works (Wang et al., 2024b; Leroy et al., 2024; Yang et al., 2025) and use a DPT head (Ranftl et al., 2021) to obtain query 3D coordinates. We observed that adding the supervision for the mapping 3D coordinates leads to more accurate query predictions. However, unlike the query 3D points, which need to exploit and capture the spatial structure of the scene,

the mapping 3D coordinates are primarily used as a supervisory signal during training. Therefore, we use a single MLP layer as the mapping head. After computing the mapping and query 3D coordinates, we multiply them by the scale factor $s$ to recover the metric scale of the scene.

### 3.3 TRAINING

The training objective follows the regression of coordinates in 3D space proposed in (Wang et al., 2024b; Leroy et al., 2024). We define the regression loss as the Euclidean distance between predicted $(X_i)$ and ground-truth $(\bar{X}_i)$ 3D coordinates as: $\ell^{\text{Reg}} = ||X_i - \bar{X}_i||$. The regression loss for the mapping head is constrained to the coordinates corresponding to the sampled features that created the map representation, while the regression loss for the query head is computed on all pixels with valid ground-truth depth values. We adopt DUSt3R's confidence-based loss, which allows the network to predict lower confidences in regions where predicting 3D coordinates might be challenging or ambiguous (*e.g.*, sky, or translucent objects). The final training objective is defined as:

$$\ell^{\text{Conf}} = \sum_{v \in \{Q, M\}} \sum_{i \in D} C_i \ell^{\text{Reg}}(v, i) - \alpha \log(C_i), \tag{2}$$

where $C_i$ is the confidence score for pixel $i$, $D$ refers either to the pixels in the query image or the map representation, and $\alpha$ is a hyper-parameter controlling the regularization (Wan et al., 2018). Please refer to Wang et al. (2024b) for further details.

## 4 EXPERIMENTS

**Absolute Pose Estimation.** At inference time, we compute the set of 2D-3D correspondences, which define 2D pixel locations in the query image $(I^Q)$ and their corresponding 3D points in the scene coordinate system defined by the map representation M. We filter correspondences with low confidence scores $(C_i < \tau)$ and set a maximum of 5,000 correspondences in PnP-RANSAC. Refer to Appendix A for FastForward's additional inference, training and datasets details.

**Competitors.** Inspired by Reloc3r (Dong et al., 2024), we group competing methods into *Seen* and *Unseen* categories. This distinction refers to whether extensive map preparation is required before a query can be localized. All methods assume that mapping images and their corresponding poses are available. While some datasets, such as Cambridge (Kendall et al., 2015), require building a SfM model to obtain these poses, others, like Wayspots (Brachmann et al., 2023), provide them in real-time via on-device tracking systems. Even with available mapping poses, *Seen* methods still require triangulating a scene (structure-based) or training a neural network (SCR). The triangulation time for structure-based approaches is dataset-specific, ranging from minutes to hours depending on the number of mapping images, whereas SCR methods can limit their training time to a few minutes. In contrast, *Unseen* methods, such as RPR, only require a curated list of nearest-neighbors for the query image. This image retrieval process can be performed very efficiently using compact image-level descriptors (Revaud et al., 2019; Arandjelovic et al., 2016), which reduces the mapping preparation time to a minimum.

### 4.1 WAYSPOTS DATASET

The Wayspots dataset (Brachmann et al., 2023) is composed of ten scenes, each with two aligned scans for mapping and localization. It contains small outdoor places of interest, such as sculptures, signs, or fountains. We compare FastForward against several state-of-the-art visual localization methods, and, as discussed, we group them into *Seen* and *Unseen* categories based on their map preparation requirements. The *Seen* group includes SCR methods like ACE (Brachmann et al., 2023) and GLACE (Wang et al., 2024a). In the *Unseen* group, we report results for Reloc3r (Dong et al., 2024) and 2D-2D feature matchers, specifically ALIKED-LightGlue (ALKD-LG) (Zhao et al., 2023; Lindenberger et al., 2023) and RoMa (Edstedt et al., 2023) paired with the E5+1 solver (Zheng & Wu, 2015; Panek et al., 2025) from PoseLib (Larsson & contributors, 2020). For ALKD-LG, we report results when extracting 1,024 features on 640px images. We omit comparisons against MASt3R since Wayspots is a subset of the Map-free (Arnold et al., 2022) training set, which was used to train it. All *Unseen* methods use the same top-20 nearest-neighbor retrieval system, which is computed in 3 seconds, requiring the extraction of image-level descriptors every 5 frames with

Table 1: **Median Pose Errors on Wayspots (Brachmann et al., 2023).** We provide the median translation and the average median rotation errors of the dataset. We also report the mapping and latency times for each method. Best results in **bold** for the *Unseen* category.

|  | $e_t$ (m) | Cubes | Bears | Winter | Insc. | Rock | Tend. | Map | Bench | Statue | Lawn | **Avg.** | $e_r$ (°) | **Latency** | **Mapping** |
|---|---|---|---|---|---|---|---|---|---|---|---|---|---|---|---|
| *Seen* | ACE | 0.05 | 0.04 | 4.76 | 0.10 | 0.03 | 1.63 | 0.07 | 0.05 | 5.50 | 1.11 | 1.33 | 9.08 | 0.1s | 5min |
| | GLACE | 0.06 | 0.03 | 5.03 | 0.10 | 0.03 | 1.69 | 0.07 | 0.06 | 5.97 | 1.30 | 1.43 | 8.87 | 0.1s | 25min |
| *Unseen* | E5+1 (ALKD-LG) | 0.09 | 0.03 | 1.17 | 0.11 | **0.03** | 0.94 | 0.08 | 0.07 | 1.47 | 1.12 | 0.51 | 7.74 | 0.8s | 3s |
| | E5+1 (RoMa) | 0.09 | **0.02** | 0.72 | **0.09** | **0.03** | 0.24 | 0.09 | 0.12 | 6.21 | **0.10** | 0.77 | 4.12 | 18.0s | 3s |
| | Reloc3r | 0.32 | 0.06 | 5.01 | 0.13 | 0.04 | 0.81 | 0.08 | 0.15 | 5.76 | 0.69 | 1.31 | 2.04 | 0.6s | 3s |
| | **FastForward** | **0.08** | 0.03 | **0.47** | 0.14 | 0.04 | **0.15** | **0.07** | 0.06 | **0.54** | **0.10** | **0.17** | **1.75** | **0.5s** | 3s |

Table 2: **Accuracy on Wayspots (Brachmann et al., 2023).** We report the accuracy under the 10cm, 10° threshold. FastForward achieves the highest number of acceptable localizations for a real-world application such as AR (Arnold et al., 2022). Best results in **bold** for the *Unseen* group.

|  | 10cm, 10°(%) | Cubes | Bears | Winter | Insc. | Rock | Tend. | Map | Bench | Statue | Lawn | **Avg.** | **Storage** |
|---|---|---|---|---|---|---|---|---|---|---|---|---|---|
| *Seen* | ACE | 95.1 | 80.0 | 0.7 | 49.7 | 100.0 | 32.9 | 55.9 | 67.8 | 0.0 | 37.0 | 51.9 | Weights |
| | GLACE | 89.6 | 86.4 | 0.0 | 47.9 | 100.0 | 37.0 | 58.3 | 64.1 | 0.0 | 40.4 | 52.4 | Weights |
| *Unseen* | E5+1 (ALKD-LG) | 52.5 | 89.1 | **2.8** | 45.7 | 96.7 | 37.1 | 54.2 | 53.8 | 0.0 | 33.2 | 46.5 | Images |
| | E5+1 (RoMa) | 53.6 | 98.3 | 0.7 | **53.3** | 99.8 | 40.8 | 56.8 | 43.1 | 0.2 | 48.4 | 49.5 | Images |
| | Reloc3r | 30.6 | 72.1 | 0.0 | 43.8 | 99.0 | 22.9 | **59.2** | 32.6 | 0.0 | 10.9 | 37.1 | Images |
| | **FastForward** | **67.8** | 94.8 | 0.4 | 31.2 | **100.0** | **41.4** | 56.8 | **70.1** | **2.7** | **48.6** | **51.4** | Images |

GeM-AP (Revaud et al., 2019). In contrast, the SCR methods, ACE and GLACE, require 5 and 25 minutes, respectively, to train their scene-specific networks. In FastForward, we sample N = 3,000 mapping features, corresponding to 20% of the total features in our map representation M.

In Table 1, we see that FastForward excels in translation estimation, reporting a median error of 0.17m, while all competitors show median errors above half a meter. Furthermore, FastForward obtains the lowest mean median rotation error. FastForward achieves these state-of-the-art results while reducing the mapping preparation times required by SCR methods and displaying the fastest localization time in the *Unseen* group. Table 2 also shows the percentage of query frames under the 10cm, 10° threshold, which determines the acceptability of an estimate for a real-world application such as AR (Arnold et al., 2022; Barroso-Laguna et al., 2024). FastForward outperforms all *Unseen* methods, including Reloc3r, which also adopts DUSt3R's architecture and was designed for visual localization. E5+1 with ALKD-LG and RoMa paired with the E5+1 solver offer strong performance in scenes with good coverage but fail when the scene presents challenging conditions, such as far-away or opposite viewpoints, *e.g.*, Winter or Lawn scenes. Additionally, RoMa's high latency (18s) is ill-suited for real-time visual localization systems; therefore, we focus our following analyses on E5+1 (ALKD-LG) given its good accuracy-latency trade-off. Regarding storage, *Unseen* methods only require images and global descriptors, while SCRs store their network weights, *e.g.*, 4MB (ACE), or 9MB (GLACE).

## 4.2 INDOOR6 DATASET

The Indoor6 dataset (Do et al., 2022) contains six indoor scenes that present challenges like repetitive or uncharacteristic areas and significant illumination changes. We also include the results of MASt3R + Kapture (Leroy et al., 2024; Humenberger et al., 2020), an approach that requires an initial SfM model and uses MASt3R as a 2D-2D matcher, and a variant of MASt3R that relies directly in its 3D point and matching heads instead of building a SfM model. Given the smaller scene area compared to the Wayspots dataset, we build a retrieval system that returns the top-10 mapping images. We use the same retrieved images for MASt3R approaches, E5+1 (ALKD-LG), Reloc3r, and FastForward. Besides, we reduce our map representation size to N = 1,500.

Table 3 (left) presents the median pose errors and the percentage of accepted query frames under the 10cm, 10° and 20cm, 20° thresholds. FastForward achieves the highest acceptance rates for both

Table 3: **Results for Indoor6 (Do et al., 2022) and RIO10 (Wald et al., 2020)**. Results on Indoor6 shows that FastForward achieves the highest accuracy among all competitors. In RIO10, MASt3R and FastForward report the best accuracies. We **bold** the best results in the *Unseen* group.

| | | Indoor6 Dataset | | | | | RIO10 Dataset | | | | | |
|---|---|---|---|---|---|---|---|---|---|---|---|---|
| | | $e_t$ (m) | $e_r$ (°) | 10cm, 10° | 20cm, 20° | Mapping | $e_t$ (m) | $e_r$ (°) | 10cm, 10° | 20cm, 20° | Mapping | Latency |
| *Seen* | MASt3R+Kapture | 0.03 | 0.5 | 89.0 | 93.6 | ∼3.5h | N/A | N/A | 24.8 | 32.6 | ∼4h | 4.5s |
| | ACE | 0.11 | 1.8 | 57.5 | 68.8 | 5min | 3.58 | 58.7 | 11.0 | 16.2 | 5min | 0.1s |
| | GLACE | 0.04 | 0.6 | 86.3 | 92.0 | 25min | 1.14 | 33.4 | 22.8 | 31.7 | 25min | 0.1s |
| *Unseen* | E5+1 (ALKD-LG) | **0.04** | **0.6** | 80.9 | 89.8 | 8s | N/A | N/A | 25.5 | 35.8 | 10s | 0.4s |
| | MASt3R | 0.13 | 0.7 | 45.9 | 76.0 | 8s | **0.17** | **5.5** | **45.1** | 58.2 | 10s | 4.5s |
| | Reloc3r | 0.09 | 0.8 | 57.4 | 72.8 | 8s | 0.47 | 9.4 | 21.4 | 32.9 | 10s | **0.3s** |
| | **FastForward** | **0.04** | **0.6** | **91.5** | **98.0** | 8s | 0.18 | **5.5** | 40.6 | **59.7** | 10s | **0.3s** |

thresholds among all competitors. FastForward surpasses even MASt3R + Kapture, a method that requires extensive mapping preparation before localization. FastForward also outperforms ACE and GLACE while reducing the mapping preparation stage to mere seconds. In the 10cm, 10° regime, FastForward boosts the accuracy of MASt3R, Reloc3r and E5+1 (ALKD-LG) by 99%, 59% and 13%, respectively, improving significantly upon other RPR approaches. Since the *Unseen* methods use the top-10 instead of top-20 retrieved images as in Wayspots, their latencies are reduced. For instance, we see that FastForward and Reloc3r localizes a new query frame in only 0.3s.

## 4.3 RIO10 Dataset

Table 3 (right) presents the results on the RIO10 dataset (Wald et al., 2020). This dataset focuses on long-term indoor localization across ten scenes with changing conditions, such as moved or replaced furniture. Since the test evaluation service only allows submissions every two weeks, we report results on the validation set.

The dynamic nature of the RIO10 dataset poses significant challenges for methods relying on structure-based representations. As a result, MASt3R + Kapture, despite being one of the best-performing methods overall, experiences significant performance degradation in these scenes. While its accuracy is slightly better than SCR approaches, its median errors could not be computed because more than half of the query pose predictions lacked sufficient correspondences for PnP-RANSAC. E5+1 (ALKD-LG) also estimates 2D-2D matches between the query and the mapping images, relying on the known structure of the scene. In some scenes, ALKD-LG failed to produce sufficient correspondences for the E5+1 solver for more than half of the query estimates; hence, as in MASt3R + Kapture, we could not compute its median pose errors. MASt3R (*Unseen*) achieves the highest 10cm, 10° accuracy, while FastForward demonstrates the best 20cm, 20° accuracy. This suggests that access to full images, as in MASt3R's approach, may be beneficial when scene conditions change, as a sparse map representation might not capture enough fine details for optimal predictions. Nevertheless, FastForward outperforms SCR methods (ACE and GLACE), E5+1 (ALKD-LG), and Reloc3r, demonstrating its robustness for long-term visual localization.

## 4.4 Cambridge Landmarks Dataset

The Cambridge dataset (Kendall et al., 2015) is an outdoor dataset consisting of six different places of interest in Cambridge. We follow recent works (Dong et al., 2024; Brachmann et al., 2023) and report results for five of these scenes. On top of previous comparisons, we compare FastForward against several classical visual localization pipelines. In the *Seen* group, we include Active Search (AS) (Sattler et al., 2016) and hLoc (Sarlin et al., 2019; 2020). In the *Unseen* group, we include several additional RPR methods (Turkoglu et al., 2021; Arnold et al., 2022; Winkelbauer et al., 2021; Dong et al., 2024). The entire retrieval index is computed in 30 seconds, requiring only the extraction of image-level descriptors with the GeM-AP global descriptor. As in the Wayspots outdoor benchmark, we use the top-20 retrieved images to compute the query localization. Moreover, we sample N = 3,000 mapping features for our map representation M. Table 4 reports the median pose errors, query latencies, and map preparation times.

Table 4: **Median Pose Errors on Cambridge Landmarks (Kendall et al., 2015)**. *Seen* methods require triangulating the scene or training a scene-specific network before being able to localize a new query image. *Unseen* methods only require a retrieval step to find the top mapping image candidates. The retrieval step can be performed for 1,000 images in under a minute (Revaud et al., 2019). We **bold** the best and underline the second best results of the *Unseen* group.

| | $e_t$ (m) / $e_r$ (°) | Great Court | King's College | Hospital | Shop Facade | Church | **Average** | **Latency** | **Mapping** |
|---|---|---|---|---|---|---|---|---|---|
| *Seen* | AS (SIFT) | 0.24 / 0.1 | 0.13 / 0.2 | 0.20 / 0.4 | 0.04 / 0.2 | 0.08 / 0.3 | 0.14 / 0.3 | 0.4s | ~35min |
| | hLoc (SP+SG) | 0.16 / 0.1 | 0.12 / 0.2 | 0.15 / 0.3 | 0.04 / 0.2 | 0.07 / 0.2 | 0.11 / 0.2 | ~1.2s | |
| | MASt3R + Kapture | 0.13 / 0.1 | 0.07 / 0.1 | 0.15 / 0.3 | 0.04 / 0.2 | 0.04 / 0.1 | 0.09 / 0.2 | 9.0s | |
| | ACE | 0.44 / 0.2 | 0.30 / 0.4 | 0.30 / 0.6 | 0.06 / 0.3 | 0.20 / 0.6 | 0.26 / 0.4 | 0.1s | 5min |
| | GLACE | 0.19 / 0.1 | 0.19 / 0.3 | 0.17 / 0.4 | 0.04 / 0.2 | 0.09 / 0.3 | 0.14 / 0.3 | 0.1s | 25min |
| *Unseen* | Relpose-GNN | 3.20 / 2.2 | 0.48 / 1.0 | 1.14 / 2.5 | 0.48 / 2.5 | 1.52 / 3.2 | 1.37 / 2.3 | N/A | ~30s |
| | Map-free | 8.40 / 4.6 | 2.44 / 2.5 | 3.73 / 5.2 | 0.97 / 3.2 | 2.91 / 5.1 | 3.69 / 4.1 | ~**0.2s** | |
| | ExReNet | 9.79 / 4.5 | 2.33 / 2.5 | 3.54 / 3.5 | 0.72 / 2.4 | 2.30 / 3.7 | 3.74 / 3.3 | ~0.4s | |
| | E5+1 (ALKD-LG) | **0.32** / **0.1** | **0.16** / **0.3** | **0.30** / 0.6 | **0.05** / **0.3** | **0.09** / **0.3** | **0.18** / **0.3** | 1.1s | |
| | MASt3R | 5.62 / 0.5 | 4.71 / 0.7 | 4.71 / 0.7 | 1.14 / 0.7 | 3.43 / 0.7 | 3.90 / 0.7 | 9.0s | |
| | Reloc3r | 0.97 / 0.6 | 0.41 / **0.3** | 0.73 / 0.6 | 0.14 / 0.6 | 0.33 / 0.6 | 0.52 / 0.5 | 0.6s | |
| | **FastForward** | 0.62 / 0.4 | 0.24 / 0.4 | 0.26 / **0.5** | 0.08 / 0.4 | 0.14 / 0.5 | 0.27 / 0.4 | 0.5s | |

ALKD-LG paired with the E5+1 solver achieves the lowest median pose errors among the *Unseen* methods. Notably, it surpasses some structure-based localizers, such as ACE, despite only requiring a retrieval step for mapping. As a method based on 2D-2D image matching, similar to AS or MASt3R + Kapture, E5+1 (ALKD-LG) relies heavily on high structural consistency and dense map coverage. Unlike Wayspots or RIO10, the Cambridge dataset offers these favorable conditions, allowing explicit matching methods to excel; however, as shown in previous sections, they struggle in more challenging or sparsely mapped environments. In Appendix C.6, we explore various E5+1 (ALKD-LG) configurations and discuss the latency-accuracy trade-offs compared to FastForward. Meanwhile, FastForward obtains the second-best median errors among the *Unseen* methods, reducing the translation error of its closest competitor, Reloc3r, by 48%. MASt3R struggles in the large-scale Cambridge scenes since they display scale ranges that are not present in its training dataset. FastForward is trained on a subset of these datasets (refer to Appendix A for details); however, its scale normalization strategy helps FastForward to generalize well to these unseen scale ranges. We extend the scale normalization discussion in Appendix C.1.

## 4.5 UNDERSTANDING FASTFORWARD

**Validation Examples.** We present qualitative results of FastForward on the validation datasets in Figure 3. For these visualizations, we use 9 mapping images and a map representation with N=1,000 features, and highlight the image regions corresponding to the selected mapping features. For training and validation, instead of using image retrieval as in the localization experiments, we randomly sample mapping images that overlap with the query image by at least 20% and not more than 85%. We use the overlapping scores provided in DUSt3R training pairs. This ensures larger scene coverage and encourages the network to learn to interpret mapping features from diverse locations. The ground-truth camera pose is shown in green for reference, while FastForward prediction is in blue. We also display the predicted 3D coordinates of the query points. Even though FastForward only uses a subset of mapping features at inference time, it still exhibits robustness comparable to DUSt3R. FastForward effectively handles repetitive patterns and symmetries by accessing only a few mapping features, demonstrating the effectiveness of our map representations.

**Runtime: Mapping vs. Querying.** Structure-based relocalizers generally offer fast query times once the scene representation is built. Consequently, structure-based methods can amortize their high mapping costs after a certain number of queries. For instance, compared to the highly efficient ACE baseline, the break-even point occurs at approximately 600 relocalizations. Thus, for high-demand locations, structure-based relocalizers become computationally more efficient in the long run. However, FastForward enables instant, on-demand relocalization for custom maps or locations where usage is unpredictable. Service providers can leverage this flexibility to offer immediate coverage, opting to build structured maps only for spots that demonstrate high popularity. Finally, FastForward allows for a configurable trade-off between runtime and accuracy by varying the number of retrieved mapping images to meet the requirements of the application (see Appendix C.2).

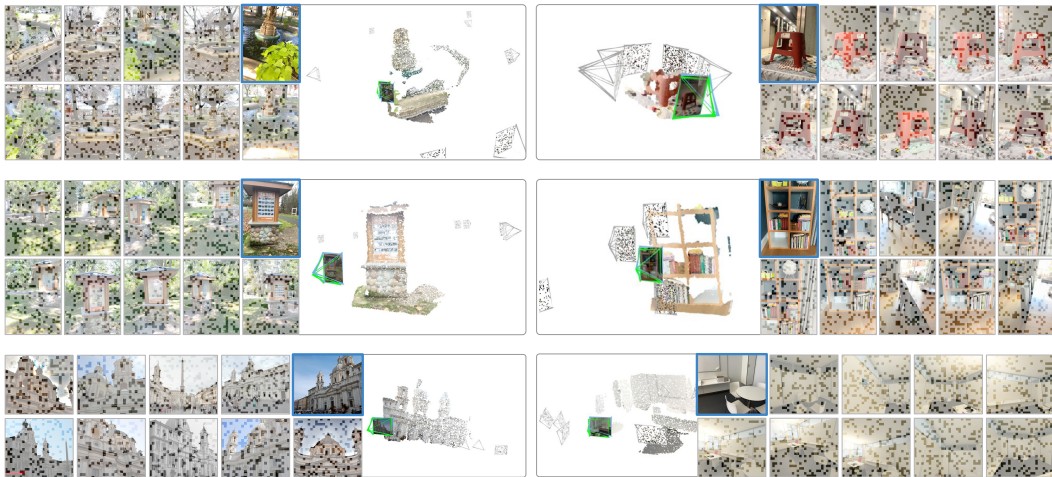

Figure 3: **Qualitative Examples**. The estimated camera pose from FastForward is shown in blue, the ground-truth pose in green, and the mapping camera poses in gray. We visualize the predicted 3D coordinates of the query points, as well as the image patches from which the mapping features are sampled. We use 9 mapping images and a map representation with N=1,000 features. FastForward effectively handles symmetries and non-discriminative patterns in the scenes. Besides, since FastForward is agnostic to the scale of the scene, it can accurately predict poses in scenes with arbitrary scales, as demonstrated in the MegaDepth (Li & Snavely, 2018) example (bottom-left).

**Limitations.** Although building a retrieval index is fast, *e.g.*, under one minute for 2,500 images using GeM-AP (Revaud et al., 2019) on a single V100 GPU, the time to extract global descriptors with a growing number of images is not negligible. In Table 7 (Appendix B.2), we investigate a version of FastForward that does not rely on image retrieval but selects reference mapping images at random or uniformly in the Wayspots dataset. This setup is much more challenging as reference images might be less relevant to the query. We observe the accuracy dropping from 51.4% to 47.8% (10cm, 10°). Other RPR methods suffer similarly, for example, Reloc3r's accuracy drops from 37.1% to 19.7% without the retrieval step. Future work could explore alternative strategies for selecting mapping images to represent the scene.

**More Details and Experiments in the Appendix.** Training and inference details are in Appendix A. Appendix B.1 reports the results in the 7-Scenes dataset (Shotton et al., 2013). Appendix B.2 shows different map representation strategies that do not require retrieval, and hence, reduce the mapping preparation time to zero. We discuss the benefits of our scale normalization in Appendix C.1. Furthermore, we study the impact of the number of mapping images and the size of the map representation N in Appendix C.2. We provide visual examples from the test set in Appendix C.4.

## 5    CONCLUSIONS

We have introduced FastForward, a method that enables fast mapping and localization through a single feed-forward pass. We have demonstrated that a visual localizer can reduce its mapping preparation requirements to a simple retrieval step and still provide state-of-the-art visual localizations. We have also shown that a sparse collection of mapping features can serve as an effective and sufficient representation of the scene for accurate visual localization. Furthermore, we have demonstrated that simple yet effective scene and scale normalization techniques can significantly improve visual localization accuracy in out-of-domain scenes. We have shown the robustness of FastForward predictions in multiple indoor and outdoor datasets, where each of them displayed unique challenges such as large-scale ranges, varying illumination conditions, and dynamic scenes. Our experiments demonstrate that we can achieve both efficient and accurate visual localization with a single feed-forward pass. FastForward outperforms state-of-the-art RPR methods on both indoor and outdoor datasets, while achieving higher accuracy to SCR methods on indoor datasets and superior or comparable performance on outdoor datasets.

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

APPENDIX

## A    TRAINING & INFERENCE DETAILS

This section provides the training parameters and datasets we used to train FastForward. Besides, we also provide some complementary inference details to those in the main paper.

**Training.** FastForward is trained on a mix of indoor and outdoor datasets. We train on a subset of the datasets used in DUSt3R/MASt3R (Wang et al., 2024b; Leroy et al., 2024), specifically: ARKitScenes (Baruch et al., 2021), WildRGBD (Xia et al., 2024), ScanNet++ (Yeshwanth et al., 2023), MegaDepth (Li & Snavely, 2018), BlenderMVS (Yao et al., 2020), and Map-free (Arnold et al., 2022) (excluding the scenes in the Wayspots dataset (Brachmann et al., 2023)).

During training, we fix the number of mapping images in M to $K = 5$, but sample varying numbers of features to create different map representation configurations such that $N \in [250, 1000]$. We initialize FastForward with the public 512-DPT weights from DUSt3R.

Only the decoder and the two heads are trained, while the encoder is frozen. We train FastForward by optimizing the loss in Equation 2 with the AdamW (Loshchilov et al., 2017) optimizer for 615k iterations. We use a batch size of 48 and a cosine learning rate scheduler with a peak learning rate of 1e-4 and a warmup of 30k iterations. We leverage float16 precision to improve GPU memory and computational efficiency. Training is performed on 8 A100-40G GPUs and completes in 5 days.

We use the overlap scores from DUSt3R (Wang et al., 2024b) and MASt3R (Leroy et al., 2024) to select the mapping images in M. We use a similar strategy to DUSt3R/MASt3R where only mapping images that overlap with the query image are valid training candidates. We set the overlapping range to $[0.2, 0.85]$. For datasets without overlapping information, *e.g.*, WildRGBD (Xia et al., 2024), we randomly sample the mapping images in M. We balance the outdoor and indoor datasets such that the model is trained with a similar number of indoor and outdoor examples.

**Inference.** At inference time, each image is resized to 512 in its largest dimension and center cropped to the closest aspect ratio used during training (Wang et al., 2024b). Before PnP-RANSAC, we filter query 3D point predictions that have a low confidence value ($C_i < \tau$, where $\tau = 1, 5$), and randomly subsample at least 5,000 correspondences. For the outdoor experiments, we use the top-20 retrieved mapping images, while the top-10 mapping images for the indoor environments.

## B    ADDITIONAL EXPERIMENTS

### B.1    7-SCENES DATASET

We present in Table 5 the median errors, accuracies, latencies, and mapping details for the 7-Scenes dataset (Shotton et al., 2013). This dataset focuses on short-term indoor localization and provides seven scenes with multiple mapping and query scans.

We observe that the *Unseen* methods perform competitively even in static scenes, where methods based on SfM localizers or SCR networks typically excel. Among the *Unseen* methods, FastForward achieves the highest acceptance rate for the 10cm, 10° threshold, while E5+1 (ALKD-LG), followed by FastForward, gets the best accuracy under the 5cm, 5° threshold. FastForward and Reloc3r obtain the lowest median translation error, while MASt3R slightly surpasses them in rotation error (-0.04°). The improvement in translation error demonstrates the benefit of having access to mapping poses at inference time, even in scenarios, *i.e.*, indoor scenes, that were represented in the training set of all *Unseen* methods. Besides the accuracy, we also report the mapping times and storage requirements. For the retrieval system of the *Unseen* methods, we apply a frame rate of fifteen in the mapping scans before building the retrieval index, *i.e.*, only one frame every fifteen is considered as a mapping candidate for the query image. The average number of images in the mapping scans is 3,700, which, after our fifteen-frame sampling, becomes 250 images. This sampling removes consecutive and redundant mapping frames and reduces the retrieval time to 7s. As discussed, the retrieval step is a much faster mapping process than those required by *Seen* methods. FastForward, as all other *Unseen* methods, needs to store the mapping images and their global descriptors for retrieval. The storage cost, therefore, depends on the number of mapping images, but is generally

Table 5: **Results on 7-Scenes dataset (Shotton et al., 2013).** We report the accuracies, median errors, and mapping preparation times for each method. FastForward achieves the highest accuracies among the *Unseen* methods. The *Unseen* methods are based on a top-10 retrieval search, and thus they can run in just a few seconds, unlike MASt3R + Kapture, GLACE, or ACE. In the Storage requirement, PC refers to Point Cloud, and Weights to the scene-specific network weights. Best results in **bold** for the *Unseen* methods group.

| | | $e_t$ / $e_r$ | 5cm, 5° | 10cm, 10° | Latency | Storage | Map Preparation | Mapping Time |
|---|---|---|---|---|---|---|---|---|
| *Seen* | MASt3R + Kapture | 0.03 / 1.06 | 73.7 | 93.5 | 4.5s | Images + PC | Point Triangulation | ∼3 hours |
| | ACE | 0.01 / 0.33 | 97.1 | 99.5 | 0.1s | Weights (4MB) | Network Training | 5min |
| | GLACE | 0.01 / 0.36 | 95.6 | 97.8 | 0.1s | Weights (9MB) | Network Training | 25min |
| *Unseen* | E5+1 (ALKD-LG) | 0.05 / 1.30 | **80.7** | 89.3 | 0.4s | Images | Retrieval | |
| | MASt3R | 0.07 / **1.01** | 26.6 | 71.9 | 4.5s | Images | Retrieval | 7s |
| | Reloc3r | **0.04** / 1.02 | 64.3 | 85.9 | **0.3s** | Images | Retrieval | |
| | **FastForward** (Ours) | **0.04** / 1.05 | 73.6 | **90.2** | **0.3s** | Images | Retrieval | |

Table 6: **Results on Wayspots dataset (Brachmann et al., 2023).** We provide the median rotation errors and the accuracy under the 10cm, 10° threshold. Additionally, we include the average median translation error and the mapping preparation time for each of the methods. ACE and GLACE train a network for each scene in Wayspots, while Reloc3r and FastForward compute a retrieval index that runs in 3 seconds for a Wayspots scene on a V100 GPU. In contrast to Reloc3r, FastForward obtains a comparable accuracy to SCR methods while reducing their mapping time. In addition, FastForward achieves the lowest rotation error. Best results in **bold** for the *Unseen* category.

| $e_r$ (°) | ACE | GLACE | E5+1 (ALKD-LG) | Reloc3r | **FastForward** |
|---|---|---|---|---|---|
| Cubes | 0.7 | 0.8 | **0.8** | 0.9 | 1.1 |
| Bears | 1.1 | 1.0 | **0.9** | 2.2 | 1.1 |
| Winter | 1.1 | 1.4 | **1.0** | 1.4 | 2.2 |
| Inscrip. | 1.6 | 1.4 | 1.2 | **1.1** | 1.9 |
| Rock | 0.8 | 0.8 | **0.8** | **0.8** | **0.8** |
| Tendrils | 36.9 | 28.9 | 23.9 | 4.4 | **3.3** |
| Map | 1.1 | 1.1 | **0.9** | 1.1 | 1.0 |
| Bench | 0.7 | 0.7 | **0.7** | 1.0 | **0.7** |
| Statue | 14.3 | 13.0 | **1.6** | 1.9 | 3.9 |
| Lawn | 32.6 | 40.2 | 45.7 | 5.6 | **1.4** |
| **Avg.** | 9.1 | 8.9 | 7.7 | 2.0 | **1.8** |
| $e_t$ (m) | 1.33 | 1.43 | 0.51 | 1.31 | **0.17** |
| 10cm, 10° (%) | 51.9 | 52.4 | 46.5 | 37.1 | **51.4** |
| Latency | 0.1s | 0.1s | 0.8s | 0.6s | **0.5s** |
| Mapping | 5min | 25min | 3s | 3s | 3s |

lower than that of classical methods that store large point clouds with high-dimensional descriptors. Mapping images and point clouds can be sub-sampled to save storage if needed; however, the storage cost of SCR methods (at least for small areas) is generally the lowest with a few MB. FastForward is competitive to even SCR when using the variant that uniformly samples mapping images instead of doing retrieval (see Table 7). In this case, only a fixed set of 20 images needs to be stored to represent an entire scene.

## B.2 WAYSPOTS DATASET

**Additional Metrics.** Table 6 shows additional metrics to the Tables 1 and 2 from the main paper. We report the median rotation errors in the Wayspots dataset (Brachmann et al., 2023). FastForward obtains the lowest rotation error among all competitors, even surpassing SCR methods while reducing their mapping preparation time from 5 or 25 minutes to a few seconds. Furthermore, as discussed in the main paper, FastForward significantly improves the median translation error, reducing the second best error from 0.51m (E5+1 with ALKD-LG) to 0.17m. This demonstrates that FastForward achieves more robust and stable localizations, particularly in challenging scenes. In the 10cm, 10° threshold, FastForward outperforms all *Unseen* methods and shows comparable accuracy to state-of-the-art SRC localizers. The Wayspots dataset uses mapping poses from real-time SLAM

Table 7: **Map Representation results on the Wayspots dataset (Brachmann et al., 2023).** We present the results of using different strategies to select the M mapping images for the map representation generation. All strategies use 20 mapping images and sample 20% of the features from each image. We also report state-of-the-art methods as a reference. Random and uniform sampling require no mapping preparation and utilize a constant map representation that can be reused for all query images, reducing the storage requirements and the localization time. Both strategies yield pose estimates with lower median translation errors than all competitors, and even outperform the accuracy of Reloc3r.

| | $e_t$ (m) | $e_r$ (°) | 10cm, 10° (%) | Mapping Time |
|---|---|---|---|---|
| ACE | 1.33 | 9.1 | 51.9 | 5min |
| GLACE | 1.43 | 8.9 | 52.4 | 25min |
| E5+1 (ALKD-LG) w/ Retrieval | 0.51 | 7.7 | 46.5 | 3s |
| Reloc3r w/ Retrieval | 1.31 | 2.0 | 37.1 | 3s |
| **FastForward** | | | | |
| Retrieval | **0.17** | **1.8** | **51.4** | 3s |
| Random | 0.31 | 2.7 | 43.9 | **0s** |
| Uniform | 0.19 | 2.3 | 47.8 | **0s** |

on the phone without any post-processing. In contrast, evaluation poses were bundle-adjusted via COLMAP (Brachmann et al., 2023). Even though mapping poses are not perfect, *e.g.*, they might suffer drift, FastForward performs very well, showing some robustness to inaccuracies in the mapping process.

**Map Representation.** Table 7 displays the results when using different strategies to select the mapping images that constitute the map representation M. The retrieval strategy selects the top-K images based on global descriptor similarity; this is the baseline approach followed in all prior experiments. We also report results for random and uniform sampling of images along the mapping scan. While retrieval-based selection is the most accurate strategy, it requires precomputing global descriptors and finding the closest mapping candidates at inference time. Random and uniform sampling strategies offer two main advantages: 1) the map representation can be computed once and reused for all query images, and 2) the mapping preparation step is eliminated since no global descriptor extraction is needed. However, the main disadvantage is that these methods are generally less accurate than the baseline retrieval strategy. Even though the retrieval system has possible limitations or failures, FastForward shows strong robustness and accuracy comparable to retrieval-free methods like SCR approaches. Moreover, our random sampling strategy simulates a retrieval failure scenario, where the system returns images unrelated to the query. Even under these conditions, FastForward surpasses Reloc3r in accuracy and achieves lower translation errors than all competitors.

## C  FASTFORWARD ANALYSES

### C.1  SCALE NORMALIZATION

We train a FastForward model without the scale normalization step detailed in Section 3.1. In this variant, we directly feed the metric translation vector to the network, allowing it to predict the 3D coordinates in the same scale as the mapping poses. Since FastForward's training directly optimizes metric 3D predictions, we remove from its training the datasets that do not provide metric ground-truth. Specifically, we remove MegaDepth (Li & Snavely, 2018) and BlenderMVS (Yao et al., 2020). In MegaDepth, the ground-truth comes from up-to-scale SfM reconstructions. BlenderMVS provides metric poses and depth maps depending on whether the images used to build the 3D models had GPS information. We follow MASt3R and treat this dataset as non-metric since not all scenes provide metric estimates. Our baseline model, *i.e.*, FastForward with the scale normalization, does not require scaled poses during training. Hence, we can augment our training set with datasets containing arbitrary scale ranges as long as they are consistent. Normalizing the translation vector within FastForward allows for more diverse and accessible training data.

Table 8: **Scale Normalization Ablation.** We show the results of FastForward when the network directly digests the mapping poses without the scale normalization proposed in Section 3.1. The scale normalization improves the accuracy as well as the generalization capability of FastForward.

| 10cm, 10° / 20cm, 20° (%) | Cambridge | Wayspots | Indoor6 | RIO10 | 7-Scenes |
|---|---|---|---|---|---|
| w/o Scale Normalization | 1.8 / 6.2 | 47.0 / 66.1 | 83.4 / 97.2 | 35.9 / 55.2 | 89.1 / 93.7 |
| **FastForward** (ours) | **26.7 / 53.6** | **51.4 / 68.7** | **91.5 / 98.0** | **40.6 / 59.7** | **90.2 / 95.8** |

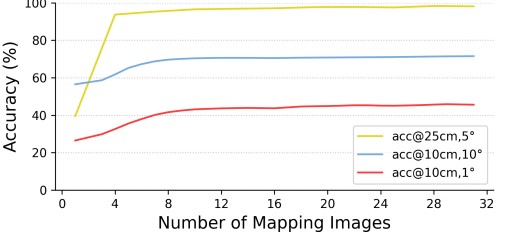 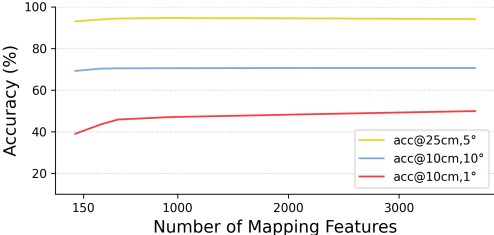

Figure 4: **Accuracy vs Number of Mapping Images**. We show the accuracy under the 10cm, 10°, 10cm, 1°, and 25cm, 5° thresholds as we increase the number of mapping images in our map representation. We fixed the size of the map representation to 768 mapping features.

Figure 5: **Accuracy vs Number of Mapping Features**. We fix the number of mapping images to 20 images and show how the accuracies change as we increase the number of mapping features that are used to create the map representation of the scene.

In Table 8, "W/o Scale Norm." refers to FastForward without the scale normalization. We observe that the scale normalization is crucial when evaluating FastForward in the Cambridge Landmarks dataset. The Cambridge dataset consist of large-scale outdoor scenes. In these scenes, the mapping images might be far from each other, and hence, the translation vectors fed into FastForward (W/o Scale Norm.) might contain larger scale ranges than those seen during training. MASt3R, even though trained with MegaDepth and BlenderMVS scenes, exhibited similar behavior in the Cambridge dataset (refer to Table 4). While performing very competitively in all indoor datasets, MASt3R's accuracy in Cambridge is only 0.5% (10cm, 10° threshold). Since FastForward has access to mapping poses at inference time, we can easily mitigate this by normalizing all translation vectors to the unit sphere (see Section 3.1). This strategy is straightforward but also very effective, *e.g.*, the 10cm, 10° accuracy in the Cambridge dataset improves from 1.8% to 26.7%. Thanks to this scale normalization, and the fact that FastForward relies on a retrieval system to turn the global pose estimation problem into a local small-scale problem, FastForward can scale to larger areas. Lastly, the results on the Wayspots and indoor datasets are comparable, with the scale-normalized version performing slightly better. This aligns with our expectations, as the scale ranges of these datasets were included in our training set.

## C.2 MAP REPRESENTATION ABLATIONS

**Number of Mapping Images.** Figure 4 presents the results when increasing the number of images that are used to create the map representation. We report the accuracy under the 10cm, 1°, 10cm, 10° and 25cm, 5° thresholds in the validation set of the Map-free dataset (Arnold et al., 2022). This experiment follows the evaluation protocol used in the Wayspots dataset. We localize the query images with respect to the mapping scan, and select the mapping image candidates using a retrieval step. Unlike the training setup, overlap information is not required for the map representation generation. We fix the map representation size to N = 768 mapping features, equivalent to sampling 100% of the features from a single mapping image. *I.e.*, the map representation size remains constant regardless of whether we use 1 or 30 mapping views. Accuracy rates for all thresholds improve with an increased number of images in the map representation, demonstrating the network's ability to incorporate multi-view information despite only a subset of the mapping features being used in the prediction. In our previous outdoor experiments, we used 20 mapping images, which we consider a good balance between accuracy and computational cost.

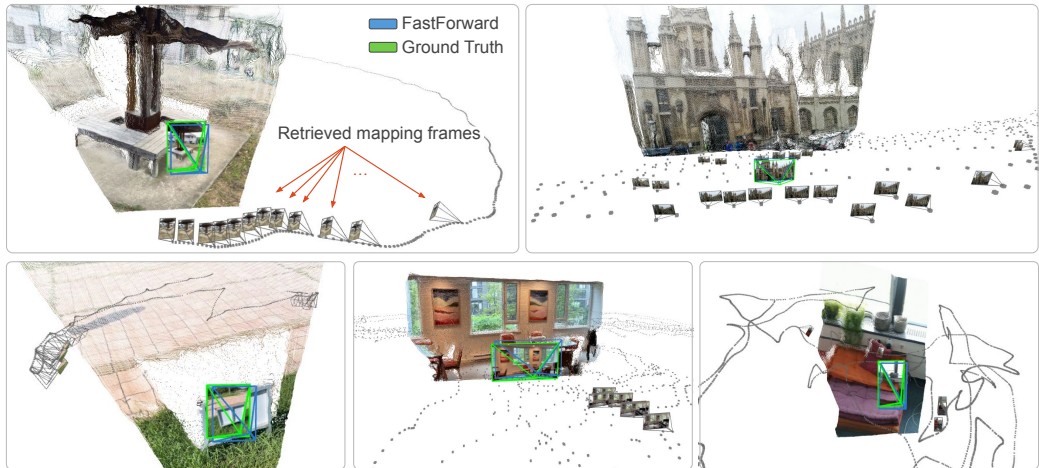

Figure 6: **Qualitative Examples.** The estimated camera pose from FastForward is shown in blue, and the ground-truth pose in green. The complete mapping scan is visualized in gray, with only the mapping images selected by our retrieval step displayed. Additionally, we visualize the predicted 3D coordinates of the query points. FastForward is able to handle symmetries, opposing viewpoints, and illumination changes. Moreover, because FastForward operates in a normalized scale space, it can handle scenes with significant scale variations, despite not being trained on them (*e.g.*, the King's College scene from the Cambridge Landmarks dataset (Kendall et al., 2015)).

**Number of Mapping Features.** Figure 5 shows the results when varying the number of mapping features used to create the map representation. All map representations are sampled from 20 mapping images. Similar to the previous ablation study, increasing the size of the map representation benefits the accuracy of FastForward. Interestingly, FastForward is affected more by the number of mapping images than by the size of the map representation itself. Accuracy under the 10cm, 10° or 25cm, 5° thresholds remain almost constant when using 150 or 3,000 mapping features. However, for finer thresholds (*e.g.*, 10cm, 1°), FastForward benefits from more mapping features. This suggesting that it can trade off accuracy on the fine threshold for reduced storage or computation.

## C.3 RUNTIME

FastForward utilizes the same feature encoder as MASt3R and Reloc3r. However, FastForward offers two key advantages: 1) Given multiple mapping images, FastForward processes a fixed set of N features in the decoder, whereas MASt3R and Reloc3r require processing all the mapping-query combinations. 2) FastForward directly provides the query 3D coordinates in the mapping scene, eliminating the need for any additional global alignment step.

FastForward extracts features from all mapping images, and hence, as in MASt3R or Reloc3r, its runtime depends on the number of mapping views. For instance, in the outdoor configuration (top-20 and $N = 3,000$), which is the most computationally expensive setup, FastForward estimates the 3D coordinates of a new query image in 0.4 seconds on a V100 GPU. Given the 2D-3D correspondences, we fed 5,000 correspondences to PnP-RANSAC (Larsson & contributors, 2020), which takes 0.1 seconds on average in a Wayspots scene to predict the pose estimate. This time could be further reduced by caching the mapping features and avoiding recomputation at inference time. Besides, FastForward could potentially use a pose head to directly predict the query pose as in Wang et al. (2025b;a) to avoid PnP. Nevertheless, FastForward provides a highly efficient solution for both mapping and localization. For example, in a Wayspots scene, retrieval takes only 3 seconds, allowing for mapping and localization of a new query in just 3.5 seconds.

## C.4 QUALITATIVE EXAMPLES

We present qualitative results of FastForward across the different test datasets in Figure 6. As previously mentioned, our map representation is constructed using 20 mapping images for outdoor

Table 9: **Pose Estimation Strategies for MASt3R (Leroy et al., 2024)**. We report the median errors and accuracy at 10cm, 10° threshold for different strategies to compute the query pose with MASt3R. In the main paper, we report the results of the default approach proposed in MASt3R for the localization tasks. Their default approach uses the matching and 3D point heads to predict the 2D-3D correspondences and PnP as the pose solver, which corresponds to the Matching - PnP entry in the table below. We provide the average time across all datasets to localize a query image for the different strategies. We also report FastForward as a reference. Best results in **bold** for the MASt3R approaches.

| | Cambridge | | Indoor6 | | RIO10 | | 7-Scenes | | |
|---|---|---|---|---|---|---|---|---|---|
| | $e_t$ / $e_r$ | Acc. | $e_t$ / $e_r$ | Acc. | $e_t$ / $e_r$ | Acc. | $e_t$ / $e_r$ | Acc. | **Time** |
| **MASt3R - Matching** | | | | | | | | | |
| PnP | 3.90 / **0.7** | **0.5** | **0.13** / **0.7** | **45.9** | **0.17** / 5.5 | **45.1** | **0.07** / **1.0** | 71.9 | 5.6 |
| Ess.Mat. + D.Scale | 4.67 / 1.0 | 0.1 | **0.13** / 0.9 | 45.8 | 0.37 / 12.4 | 29.6 | **0.07** / **1.0** | **72.3** | 19.4 |
| **MASt3R - Direct Reg** | | | | | | | | | |
| PnP | 4.01 / 0.9 | 0.2 | **0.13** / **0.7** | 43.8 | 0.21 / **5.4** | 35.1 | 0.08 / 1.2 | 69.2 | **4.7** |
| Ess.Mat. + D.Scale | **3.87** / 0.9 | 0.2 | **0.13** / 0.9 | 45.8 | 0.29 / 9.5 | 29.6 | 0.08 / 1.1 | 69.8 | 13.1 |
| **FastForward** | 0.27 / 0.4 | 26.7 | 0.04 / 0.6 | 91.5 | 0.18 / 5.5 | 40.6 | 0.04 / 1.1 | 90.2 | 0.4 |

scenes and 10 for indoor scenes, with 20% of the features sampled from each image. The ground-truth camera pose is shown in green, and FastForward's in blue. The mapping scan trajectory is shown in gray, and only the mapping images selected by the retrieval step are visualized. We also display the predicted 3D coordinates of the query points. We observe that accessing only a subset of mapping features is sufficient for robust localization, even in challenging scenarios such as scenes with significant illumination variations, repetitive patterns (e.g., white walls), symmetric objects, or opposing viewpoints. Furthermore, FastForward can handle large-scale scenes, such as those in Cambridge, despite being trained on outdoor data limited to Map-free (Arnold et al., 2022), MegaDepth (Li & Snavely, 2018), and BlenderMVS (Yao et al., 2020) datasets, which present small to mid-scale ranges (Map-free) or arbitrary scales (MegaDepth / BlenderMVS). Moreover, in addition to the robustness against unseen scale ranges, FastForward demonstrates outstanding performance on some traditional challenges, such as opposing shots. For example, the bottom-left image from the Wayspots dataset (Lawn) illustrates that FastForward is able to estimate an accurate pose even though the mapping scan was taken from an opposing viewpoint.

## C.5 POSE ESTIMATION WITH MASt3R

FastForward directly predicts the query scene coordinates to establish 2D-3D correspondences, enabling pose estimation with the PnP solver (Gao et al., 2003). In contrast, MASt3R provides a descriptor head for estimating the keypoint matches between the image pairs, thus supporting various correspondence estimation methods and pose solvers. In all previous experiments, we reported MASt3R's results using its default visual localization pipeline, which employs the PnP solver and 2D-3D correspondences derived from its matching and 3D point heads.

One alternative to PnP is to estimate the 2D-2D correspondences via only the matching head and then compute the Essential matrix. Since the Essential matrix is up to scale, the predicted depth maps from the 3D point head are used to recover the metric scale. We refer to this approach as Ess.Mat + D.Scale in Table 9. For more details, we refer to Leroy et al. (2024) and Arnold et al. (2022). Besides, MASt3R is also able to compute correspondences directly from the predicted point cloud, similar to DUSt3R (Wang et al., 2024b), without using the matching head. We refer to this approach as direct regression (Direct Reg). The direct regression approach can be paired with either the PnP or the Essential matrix solver.

As shown in Table 9, the PnP solver performs comparably to the Essential matrix solver, even without relying on the ground-truth camera calibration of the reference view. However, a key distinction between the solvers is the computational efficiency. The Essential matrix solver requires solving for the essential matrix for each mapping image, significantly increasing localization time compared to the single-run PnP approach. Furthermore, while the direct approach performs well on some

Table 10: **Pose Estimation Ablations for ALIKED-LG (Zhao et al., 2023; Lindenberger et al., 2023) with the E5+1 solver (Zheng & Wu, 2015)**. We report the median errors, accuracy at 10cm, 10° threshold, and the latencies for the different feature extractor and RANSAC configurations. In the main paper, we report the results with 1,024 keypoints and 1,000 maximum RANSAC iterations as our baseline configuration. We also report FastForward as a reference. Best results in **bold** for the E5+1 (ALKD-LG) configurations.

| E5+1 (ALKD-LG) | | Wayspots Dataset | | | | Cambridge Dataset | | | |
|---|---|---|---|---|---|---|---|---|---|
| Num. Kpts | RANSAC | $e_t$ (m) | $e_r$ (°) | 10cm, 10° | Latency (s) | $e_t$ (m) | $e_r$ (°) | 10cm, 10° | Latency (s) |
| 1,024 | 1,000 | **0.51** | **7.7** | **46.5** | 0.77 | **0.18** | **0.3** | 37.6 | 1.10 |
| 512 | 1,000 | 0.69 | 9.2 | 44.3 | 0.63 | 0.19 | **0.3** | **38.0** | 0.90 |
| 256 | 1,000 | 1.57 | 17.4 | 37.9 | 0.58 | 0.21 | 0.4 | 35.9 | 0.73 |
| 128 | 1,000 | 2.70 | 24.6 | 27.8 | 0.56 | 0.23 | 0.4 | 35.0 | 0.63 |
| 1,024 | 500 | 0.52 | 8.3 | **46.5** | 0.66 | 0.23 | 0.4 | 37.6 | 0.82 |
| 1,024 | 100 | 0.68 | 9.6 | 44.2 | 0.57 | 0.23 | 0.4 | 37.5 | 0.76 |
| 64 | 100 | 5.72 | 56.0 | 15.6 | **0.54** | 0.31 | 0.5 | 26.7 | **0.54** |
| **FastForward** | | 0.17 | 1.8 | 51.4 | 0.49 | 0.27 | 0.4 | 26.8 | 0.49 |

datasets, it fails on more challenging scenes, particularly those with dynamic elements, such as in the RIO10 dataset.

## C.6 POSE ESTIMATION WITH THE E5+1 SOLVER

As discussed in the main paper, the E5+1 solver recovers the absolute pose from 2D-2D correspondences between the query and two or more mapping images. In Table 10, we study the trade-offs between accuracy and latency when pairing the E5+1 solver with the ALIKED-LightGlue (ALKD-LG) feature matcher. Specifically, we vary the number of keypoints extracted by ALIKED and the maximum number of RANSAC iterations in the E5+1 solver, analyzing how different configurations impact performance compared to our baseline configuration (1,024 keypoints and 1,000 iterations).

We observe that in well-structured scenes like Cambridge, just a few keypoints suffice for accurate pose estimation. Furthermore, latency can be improved by reducing the number of RANSAC iterations. Nevertheless, FastForward offers competitive results in Cambridge while remaining faster. We evaluated a lightweight configuration (64 keypoints and 100 RANSAC iterations) to test the latency limits of E5+1 (ALKD-LG); however, this configuration reports higher errors and latency (0.54s) compared to FastForward (0.49s). In contrast, on the Wayspots dataset, reducing the number of keypoints significantly degrades performance. Wayspots contains challenging scenes where 2D-2D matchers may struggle to find stable structures; consequently, optimizing for latency severely impacts the accuracy of the E5+1 (ALKD-LG) approach. In Wayspots, FastForward offers faster and more accurate pose estimates than any of the proposed E5+1 (ALKD-LG) configurations.

