# OpenReview forum: "A Scene is Worth a Thousand Features: Feed-Forward Camera Localization from a Collection of Image Features"
_ICLR.cc/2026/Conference — ICLR 2026 Poster_

### Official Review · Reviewer_yDpi · 2025-10-28

**Soundness:** 3
**Presentation:** 3
**Contribution:** 3
**Rating:** 6
**Confidence:** 4

**Summary:**

The paper introduces a method for visual mapping and localization. The development of this method is motivated by two observations: the recent success of vision models, such as DUSt3R, in generating 3D models and depth information from multiple images of a scene, and the comparatively long time current localization methods require to construct a scene representation that facilitates accurate, robust, and generalizable mapping.

The core of the method involves training a model to predict the 3D coordinates for pixels in a query image. During training, the model processes query features and sampled features derived from a subset of posed reference images. Each reference mapping feature is augmented with a ray embedding, which encodes the corresponding camera’s position and viewing direction. These augmented reference features are then matched with the query features using an attention mechanism to regress the 3D coordinates of the 2D query pixels. For inference, a small set of reference images is first retrieved from a database (using either an image retrieval process or random sampling). These retrieved images are processed alongside the query image to regress the 3D coordinates of query pixels. The resulting 2D-3D correspondences are subsequently utilized to estimate the query image's absolute pose via a PnP RANSAC procedure.

The proposed method is trained on a collection of indoor and outdoor datasets. It is evaluated on a separate collection of small- to medium-scale indoor and outdoor localization benchmarks, specifically the Cambridge Landmarks, Wayspots, and Rio10 datasets. The method is compared against existing approaches categorized by how they represent the target scene (the mapping process): (a) methods which use Structure-from-Motion (SfM) to explicitly represent the scene as a 3D model (b) Scene Coordinate Regression (SCR) methods, which encode the scene geometry implicitly within the model weights (require per-scene training) and (c) Relative Pose Regression (RPR) methods, which represent the scene via image encodings and their camera poses followed by direct regression of the relative pose between one or more fetched nearest neighbors (minimal time for mapping).

**Strengths:**

1. The method presents a novel approach to visual mapping and localization, which enables fast mapping time and the demonstrated ability to generalize to scenes not seen during training.
2. The proposed method outperforms recent state-of-the-art RPR methods (minimal mapping time) and outperform SCR methods on several scenes from the Wayspots dataset
3. The paper provides a clear and well-articulated motivation for the proposed architecture, grounding the work in the limitations of existing methods and the capabilities of modern 3D vision models.
4. The paper is well structured and provides ablation study examining the impact of scale normalization and map size (number of reference images/features), providing insight into the method's core components and hyperparameter sensitivity.

**Weaknesses:**

1. The evaluation lacks a comprehensive discussion of the runtime and storage tradeoffs relative to other baseline methods. Specifically, the proposed approach relies on local feature matching and extraction, which introduces an overhead either at inference (runtime) or in storage (pre-computed features). This is a critical comparison point against RPR methods, which typically only require global feature matching, and against SCR methods. While SCR may incur a long initial training/mapping time, its inference time is generally very fast, making the accuracy/speed trade-off unclear. The paper also lacks the discussion of the one-off mapping effort versus the repeated query time to fully contextualize the value proposition of the proposed technique across all phases (training/mapping/querying).

2. The evaluation is confined to small- to medium-scale scenes. Consequently, the method's scalability to larger, city-scale environments is not demonstrated. As structure-based methods typically show strong performance and robustness in such large-scale settings this comparative assessment (or at least discussion) is necessary.

**Questions:**

1. Could the authors please provide a comparative table and discussion detailing the trade-offs across all baselines regarding runtime (mapping and querying), storage requirements, and localization accuracy?

2. What are the authors' specific assumptions and engineering considerations required for scaling this method to significantly larger, kilometer-scale environments?

---

> ### Author Response · Authors · 2025-11-20
>
> We thank the reviewer for their time and effort.
>
> *“**1.** Could the authors please provide a comparative table and discussion detailing the trade-offs across all baselines regarding runtime (mapping and querying), storage requirements, and localization accuracy?”*
>
> | Cambridge | Mapping (s) $\downarrow$ | Querying (s) $\downarrow$ | Acc. 10cm, 10° $\uparrow$ | Acc. at 25cm, 2° $\uparrow$ | Storage |
> |:---:|:---:|:---:|:---:|:---:|:---:|
> | Kapute + MASt3R | ~2100 | ~9.0 | 61.6 | 83.8 | Images + Point cloud |
> | ACE | 300 | 0.1 | 26.3 | - | Network weights (5Mb) |
> | E5+1 (ALIKED-128kpts) | 30 | 2.2 | 32.5 | 60.7 | Images |
> | E5+1 (RoMa) | 30 | ~20.0 | - | 67.5 | Images |
> | MASt3R | 30 | ~9.0 | 0.5 | 1.6 | Images |
> | Reloc3r | 30 | 0.6 | 11.3 | 36.4 | Images |
> | FastForward (Paper) | 30 | 2.8 | 26.1 | 60.1 | Images |
> | FastForward (5k Pts) | 30 | 0.5 | 26.7 | 60.8 | Images |
>
> FastForward provides a good trade-off between accuracy and latency. Just by randomly subsampling the 2D-3D correspondences before PnP-RANSAC, we can get FastForward localization time under half a second without hurting its accuracy. FastForward even provides faster localization times than Reloc3r, which was designed to overcome the limitation of using DUSt3R/MASt3R for visual localization. In contrast, methods that outperform FastForward, eg, Kapture+MASt3R, or E5+1 w/ RoMa, take 9s and 20s to localize, which are arguably not suited for real-time applications. Note also that FastForward does not require any expensive mapping requirements, reducing the triangulation (2100s) or training network (300s) steps to mere retrieval indexing (30s). We believe that FastForward is a step forward for structure-free localization methods.
>
> We also provide results for the Wayspots dataset:
> | Wayspots | Mapping (s) $\downarrow$ | Querying (s) $\downarrow$ | Acc. 10cm, 10° $\uparrow$ | Trans. err. (m) $\downarrow$ | Storage |
> |:---:|:---:|:---:|:---:|:---:|:---:|
> | ACE | 300 | 0.1 | 51.9 | 1.33 | Network weights (5Mb) |
> | GLACE | 1500 | 0.1 | 52.4 | 1.43 | Network weights (25Mb) |
> | Reloc3r | 3 | 0.6 | 37.1 | 1.31 | Images |
> | E5+1 (ALIKED-2048) | 3 | 1.4 | 45.9 | 0.41 | Images |
> | E5+1 (RoMa) | 3 | 19.4 | 47.3 | 0.77 | Images |
> | FastForward (Paper) | 3 | 2.8 | 51.0 | 0.17 | Images |
> | FastForward (5k Pts) | 3 | 0.5 | 51.4 | 0.17 | Images |
>
> FastForward matches the accuracy of scene coordinate regressor networks, ie, ACE or GLACE, while reducing its mapping preparation time significantly (300s or 1500s to just 3s). Besides, Wayspots dataset contains some specific scenes with extreme viewpoint differences, eg, Tendrils, Statue, or Lawn (see Table 3 main paper). In such scenes, FastForward is able to deal with the challenging scenarios, eg, opposing shots or symmetric scenes. We can observe this by looking at the median translation errors, which hints that FastForward is more robust than competitors when facing such extreme examples, achieving the state-of-the-art translation error in this dataset (0.17m).

---

> ### Author Response · Authors · 2025-11-20
>
> **Discussion:**
>
> **Runtime (mapping and querying):** We see considerable opportunity to improve the runtime time of FastForward. Still, we can assume that its relocalization time is generally higher than that of structure-based relocalizers that build a scene representation. This means, after a certain number of relocalizations, structure-based methods will have amortized their mapping costs. For the highly efficient ACE baseline, the strike-even point is at 600 relocalizations. At locations with high demand, structure-based relocalizers will be computationally more efficient in the long run. However, for positioning service providers it is sometimes difficult to predict which locations will incur high usage, especially if users are allowed to create their own maps, even in rural areas. FastForward allows providers to enable instant, on-demand relocalization for custom made maps. The provider can still choose to build structured maps for those particular spots that prove to be popular.
>
> **Storage:** Like most RPR methods (all Unseen methods in Table 1), FastForward needs to store the mapping images and their global descriptors for retrieval. Nothing more is required. The storage cost therefore depends on the number of mapping images, but is generally lower than that of classical methods that store large point clouds with high-dimensional descriptors. Point clouds can be sub-sampled to save storage, but so can the mapping images. The storage cost of SCR methods (at least for small areas) is generally the lowest with a few MB. FastForward is competitive to even SCR when using the variant that uniformly samples mapping images instead of doing retrieval (see Supp. B.2, Table 7). In this case, only a fixed set of 20 images need to be stored to represent an entire scene.
>
> **Accuracy:** For scenes where the mapping coverage is good (i.e., viewpoint changes between mapping and query are moderate), structure-based relocalizers are more accurate than FastForward, see e.g. the Cambridge dataset. For scenes with extreme viewpoint changes, like front-to-back relocalization, FastForward outperforms some structure-based relocalizers, see results on the Wayspots dataset (Table 3). Also see some of the qualitative videos in the supplement (e.g., wayspots_lawn.mp4). For SCR, extreme viewpoint changes have been addressed in a recently-released ICCV25 paper (ACE-G [1]) but compared to even this latest work, FastForward is competitive on Indoor6 and outperforms ACE-G on Cambridge and RIO10.
>
> **Conclusion (value proposition of FastForward):** FastForward enables instant, on-demand relocalization which is particularly useful for user-generated maps with unclear usage expectations. Additionally, if mapping coverage is only partial (e.g., when non-expert users record mapping images), FastForward offers reasonable accuracy even under extreme viewpoint changes between query and mapping images.
>
> In terms of training, FastForward (starting from pre-trained DUSt3R) can be recreated with a one-time cost of ~$3000 which is neglectable for commercial entities (training completes in 5 days on 8 A100-40Gb GPUs). Most modern pipelines come with some sort of pre-trained components: matchers (LightGlue), features (RoMa) or encoders (ACE) and their one-time training costs are usually not considered a limitation.
>
> *"**2.** What are the authors' specific assumptions and engineering considerations required for scaling this method to significantly larger, kilometer-scale environments?”*
>
> In terms of engineering effort required to scale FastForward to large-scale environments, we do not foresee significant challenges. Similar to other scalable hierarchical systems, like hLoc, FastForward relies on image retrieval to get the relevant reference images for the current query, turning the global pose estimation problem into a small-scale problem that can be solved locally. In practice, many more options for improvements are available depending on the application, like using GPS as a prior to select reference images.
>
> For existing large-scale benchmarks, like Aachen Day-Night, we expect structure-based relocalizers to outperform FastForward because these are highly structured environments with good mapping coverage. In this regard, Aachen Day-Night is similar to Cambridge. Not much public training data (with scale-metric poses) exists for these types of scenes. (Still FastForward performs competitively in Cambridge).
>
> [1] Bruns et al., ACE-G: Improving Generalization of Scene Coordinate Regression Through Query Pre-Training, ICCV 2025.

---

> > ### Author Response · Authors · 2025-11-24
> >
> > We noticed an error in the latency calculation for the E5+1 baseline suggested by Reviewer nKKy in our previous comment and sincerely apologize for this oversight. The initial reported timings for the E5+1 baseline included the image loading and preparation overhead. We have corrected the timing below:
> > | Cambridge | Mapping (s) $\downarrow$ | Querying (s) $\downarrow$ | Acc. 10cm, 10° $\uparrow$ | Acc. at 25cm, 2° $\uparrow$ | Storage |
> > |:---:|:---:|:---:|:---:|:---:|:---:|
> > | Kapute + MASt3R | ~2100 | ~9.0 | 61.6 | 83.8 | Images + Point cloud |
> > | ACE | 300 | 0.1 | 26.3 | - | Network weights (5Mb) |
> > | E5+1 (ALIKED-128kpts) | 30 | **0.7** | 32.5 | 60.7 | Images |
> > | E5+1 (RoMa) | 30 | **17.5** | - | 67.5 | Images |
> > | MASt3R | 30 | ~9.0 | 0.5 | 1.6 | Images |
> > | Reloc3r | 30 | 0.6 | 11.3 | 36.4 | Images |
> > | FastForward (Paper) | 30 | 2.8 | 26.1 | 60.1 | Images |
> > | FastForward (5k Pts) | 30 | 0.5 | 26.7 | 60.8 | Images |
> >
> > | Wayspots | Mapping (s) $\downarrow$ | Querying (s) $\downarrow$ | Acc. 10cm, 10° $\uparrow$ | Trans. err. (m) $\downarrow$ | Storage |
> > |:---:|:---:|:---:|:---:|:---:|:---:|
> > | ACE | 300 | 0.1 | 51.9 | 1.33 | Network weights (5Mb) |
> > | GLACE | 1500 | 0.1 | 52.4 | 1.43 | Network weights (25Mb) |
> > | Reloc3r | 3 | 0.6 | 37.1 | 1.31 | Images |
> > | E5+1 (ALIKED-2048) | 3 | **1.1** | 45.9 | 0.41 | Images |
> > | E5+1 (RoMa) | 3 | **18.0** | 47.3 | 0.77 | Images |
> > | FastForward (Paper) | 3 | 2.8 | 51.0 | 0.17 | Images |
> > | FastForward (5k Pts) | 3 | 0.5 | 51.4 | 0.17 | Images |

---

> > > ### Comment · Reviewer_yDpi · 2025-11-28
> > >
> > > I thank the authors for their thorough response and for addressing my questions. Based on this response and the extended correspondence with reviewer nKKy, I will keep my positive rating, and am inclined to raise it.

---

> > > > ### Author Response · Authors · 2025-12-03
> > > > **Updates in the Post-Rebuttal Version**
> > > >
> > > > We thank Reviewer yDpi for their constructive feedback. We have uploaded a revised version of the paper incorporating the rebuttal discussions. Specifically:
> > > > - We added a discussion on the trade-off between mapping and querying times in Section 4.5 (Runtime: Mapping vs. Querying), specifying the advantages of FastForward compared to efficient pose estimators like ACE.
> > > > - We included and discussed latency metrics across all benchmarks.
> > > > - We added storage requirements to Table 2 and expanded the storage discussion in Appendix B.1.
> > > > - We extended the discussions about FastForward accuracy compared to other structure-based methods in Sections 4.1-4.4.
> > > > - As mentioned in our response to Reviewer Ryg4, we included a discussion regarding scaling FastForward to large areas in Appendix C.1.

---

### Official Review · Reviewer_13TE · 2025-10-30

**Soundness:** 2
**Presentation:** 3
**Contribution:** 2
**Rating:** 2
**Confidence:** 5

**Summary:**

This paper addresses scene coordinate regression for camera relocalization without per-scene model training or reconstruction. To this end, the authors first retrieve a set of mapping images with known poses, then extract and randomly sample ViT features from the mapping images, and finally fuse the mapping features with query features to predict the scene-coordinate map of the query image. The camera pose of the query image is recovered via PnP between the predicted scene-coordinate map and the 2D image coordinates. To demonstrate efficiency, the authors conduct experiments on multiple datasets and achieve results comparable to model-based methods, while being substantially more efficient due to the reconstruction-free design.

**Strengths:**

* The paper is well-motivated; building and storing maps is expensive, especially for large-scale scenes. Therefore, this work provides an alternative solution that could address this limitation.
* The method maximizes reuse of existing public pretrained weights (DUSt3R), which makes reproduction and extension easier.
* The additional reconstruction head on mapping images is reasonable and offers additional supervision during training.
* The experiments are extensive and cover various scene scales and conditions.

**Weaknesses:**

* The contribution of this paper is limited:

  * First, in terms of problem formulation, the paper still works on scene coordinate regression. Compared with previous scene-agnostic works, the main change is that only the poses are given for the mapping images, while the scene structure is reconstructed implicitly during inference.
  * Compared with DUSt3R, the paper extends the encoder to handle multiple images with known poses by adding additional ray encodings to the tokens.
  * The scene normalization is hard to claim as a contribution: aligning camera poses to an anchor keyframe is common practice in 3D vision; scale normalization is also fairly trivial since SfM reconstructions are scale-normalized by default, so scenes are effectively normalized—especially for model-based camera relocalization.

* The motivation is to make camera localization more efficient; however, the method is still based on scene coordinate regression + PnP. As a result, per-query PnP still dominates computation. In contrast, 3D model-based methods only need to reconstruct the scene once, and the per-query cost of retrieval and PnP is low. Therefore, it is unclear whether the proposed solution is truly more efficient in terms of marginal cost, especially considering the training time.

* The solution still relies on image retrieval, but the paper does not discuss the limitations and failure cases of retrieval—an area where scene-specific localization models can be advantageous. The authors do investigate uniformly sampling the mapping images; however, this alternative provides little new information since it is generally inferior and arguably invalid for the stated setting.

**Questions:**

- First, please refer to the weaknesses.
- Second, how the proposed solution is robust to inaccurate camera pose of the mapping images?

---

> ### Author Response · Authors · 2025-11-20
>
> We thank the reviewer for their time and effort.
>
> *“The contribution of this paper is limited”*
>
> While each individual contribution might seem obvious in hind-sight, the resulting system is unique and outperforms its competitors. Reviewer nKKy mentioned the recent arXiv report “A Guide to Structureless Visual Localization” (Panek et al.)  in which the authors try to adopt MASt3R in a visual relocalization pipeline. They state: “the MASt3R implementation released by the authors of [65] cannot make use of known intrinsics and camera poses of the database images. However, to the best of our knowledge, the same limitation applies to all other 3D
> reconstruction approaches based on the relative pose regression [132, 129, 40]. In all cases, adjusting the implementation is highly non-trivial”. FastForward fills this gap. The system closest to FastForward is Reloc3r which we outperform across all datasets by a significant margin.
>
>
> *“how the proposed solution is robust to inaccurate camera pose of the mapping images?”*
>
> FastForward is robust to some extent. For example, the Wayspots dataset uses mapping poses from real-time SLAM on the phone without any post-processing. (In contrast, evaluation poses were bundle-adjusted via COLMAP. See the ACE paper for details.) Even though mapping poses are not perfect, e.g., they might suffer drift, FastForward performs very well on Wayspots.
>
>
> *“however, the method is still based on scene coordinate regression + PnP. As a result, per-query PnP still dominates computation.”*
>
> We did not tune the runtime of FastForward as much as we should have, passing all correspondences of the DPT head to RANSAC+PnP which resulted in an unnecessary amount of computation. After batching some network operations, and randomly subsampling the final correspondences before RANSAC+PnP, we can report an end-to-end relocalization time of 500ms (same accuracy). For a discussion of the tradeoff between mapping and relocalization time, please see our response to reviewer yDpi.
>
>
> *“The solution still relies on image retrieval, but the paper does not discuss the limitations and failure cases of retrieval”*
>
> We apologize, but we are unsure which kind of discussion or experiment the reviewer would like to see. We validate our approach on various datasets and any short-comings of retrieval are baked into our results when we compare to scene-specific methods without retrieval. It is also not clear to us why the reviewer thinks that our experiments with random sampling of reference views are “invalid” - it is essentially simulating broken retrieval which seems very relevant to the question here. The results slightly degrade but not by a large margin. In particular, FastForward without retrieval still outperforms Reloc3R with retrieval, and the ranking of methods does not change (see Table 7 in the supplement).

---

> > ### Author Response · Authors · 2025-12-03
> > **Updates in the Post-Rebuttal Version**
> >
> > We thank Reviewer 13TE for their constructive feedback. We have uploaded a revised version of the paper addressing the reviewer's suggestions. Specifically:
> > - We added a discussion regarding potential inaccuracies in mapping poses and demonstrated how FastForward is robust to them (see Appendix B.2, Additional Metrics).
> > - We extended the discussion on our retrieval ablation (Appendix B.2, Map Representation). We explain how random sampling simulates a retrieval failure scenario and show that, even under these conditions, FastForward outperforms its closest competitor, Reloc3r.
> > - We addressed the comments regarding PnP runtime in Appendix C.3 and added a new discussion on the trade-off between fast mapping and fast querying methods in Section 4.5 (Runtime: Mapping vs. Querying).

---

### Official Review · Reviewer_Ryg4 · 2025-10-31

**Soundness:** 3
**Presentation:** 3
**Contribution:** 3
**Rating:** 8
**Confidence:** 4

**Summary:**

This article presents a novel visual localization method called FeedForward, which overcomes some of the difficulties associated with traditional mapping approaches. Unlike conventional methods that build complete 3D models or train scene-specific networks, FeedForward uses a limited set of image features embedded in 3D space to estimate camera positions in a single feed-forward pass. The core idea is to represent multiple mapping images as a collection of features embedded in 3D space and use these mapping features to predict image-to-scene correspondences for the query image. It is show with various experiments that the method offers significant advantages in terms of efficiency, scalability and strong generalization capabilities.

**Strengths:**

Although the architectural design is inspired by DUSt3R, the idea of ​​taking a random sample of features from  multiple posed mapping images instead of considering a reference image as input to the encoder-decoder frame to predict accurate 3D coordinates of the query directly into the map's coordinate system is original and rather clever.

The authors demonstrate that a scene representation consisting of only a few hundreds mapping features obtained with their FeedForward method  is sufficient for accurate visual localization, where building such map is magnitude faster than using competitive methods (including traditional SfM, SCR methods or even Geometric Foundation models such as MAST3R) without  significantly drop in localization accuracy.

The authors employ a simple yet effective technique for generalizing to new domains by defining one reference mapping image and transforming all other images such that the scene is centered at the origin of the coordinate system. The method computes the scene scale as the maximum camera translation in any spatial coordinate after normalization, which enables metric recovery from normalized predictions using the computed scale factor. This technique allows, on the one hand,  to exploit both metric and non-metric training data and on the other hand to enhances the network's ability to adapt to different scale ranges.

The model was tested both on classical localization datasets such as Cambridge Landmark, Indoor6, RIO10, 7-Scenes and the more challenging extreme localization dataset Wayspot.

The article is well-written, well-structured, and easy to follow. I also appreciate that the author's added a section on the model's limitations.

**Weaknesses:**

The model was tested on various standard datasets such as Cambridge Landmark, Indoor6, RIO10, and 7-Scenes, which are relatively small-scale compared to the scene size. Although these tests provide valuable insights into the model's performance in a variety of settings, it remains unclear how the model would fare on larger-scale datasets like Aachen Day-Night and InLoc compared to e.g MASt3R (Section 4.4 of the corresponding paper) that present more complex and challenging appearance variations.  Evaluating the model's robustness and adaptability to such scenarios would be crucial to better understanding of the model's strengths and limitations in real-world deployment.


At inference time, the model estimates camera poses using PnP-RANSAC with 2D-3D correspondences, which involves an extra optimization step. Given that the query head generates 3D coordinates for each 2D point, it would be interesting to evaluate the accuracy of the pointmap (either evaluating the corresponding depth-map or the  3D reconstruction) directly, in comparison with other Geometric Foundation Model (GFM) methods, the latter ones also evaluated on a per-image basis. To compare to models relying on a pair of images, FeedForward could consider a single image as map, but a denser feature sets, while models predicting pointmaps jointly for multiple views (e.g VGGT) a leave-one-out approach could be employed to construct the map for FeedForward.

**Questions:**

In the MAST3R/DUST3R paper, localization results are presented for the validation and test sets of the Map-free dataset. Although Figure 4 (supplementary) includes graphs illustrating accuracy as a function of the number of map images in the validation set, a fair comparison with other methods is difficult due to differences in thresholding compared to those used in Evaluation Leaderboard of the Map-free Visual Relocalization. To better assess the model's performance in this context, it is strongly recommended to also provide results obtained using the same evaluation protocol for single view and e.g. top 20.

---

> ### Author Response · Authors · 2025-11-20
>
> We thank the reviewer for their time and effort.
>
> We agree that reporting a metric used in the evaluation protocol of the Map-free dataset would help to compare FastForward against other methods. We report below the precision for the 25cm and 5° threshold, which is one of the main metrics in the Mapfree leaderboard. As in Figure 4 in the supplement, we compute the precision when FastForward has access to a different number of mapping frames:
> |Mapfree Validation | Top1 | Top5 | Top10 | Top15 | Top20 |
> |:---:|:---:|:---:|:---:|:---:|:---:|
> | FastForward | 39.5% | 95.3% | 97.1% | 97.8% | 98.1% |
>
> The accuracy increases as more mapping images are fed into the model. As in our ablation, we sample a constant number of mapping tokens, such that the map representation is constant regardless of the number of mapping images. Although FastForward has not been trained with only 1 mapping image, it shows that in that scenario still can provide useful localizations. We see that as we add more mapping images, the accuracy improves significantly. Besides, we also observe that the accuracy saturates at the top10 mapping images mark. As discussed in the supplement, FastForward can trade compute/latency for accuracy under strict thresholds (see Supp. C.2 for more details).
>
> We would like to note that these numbers are not directly comparable to the official validation set of Mapfree. The validation set of Mapfree only provides a single mapping image, and a full query scan. Since FastForward studies the localization of a query image wrt a collection of mapping images, we split the official Map-free training set (where full scans are provided) into our own training and validation sub-sets. The reported results in the previous table correspond to our validation set. Just to clarify, this validation is small, and we do not train on it. Our validation consists of 5 scenes, being smaller than the validation used in DUSt3R/MASt3R (65 scenes). Our goal with this experiment was to display the impact of adding mapping images. Therefore, even though both validation sets are not directly comparable, we believe that our ablation study gives a good signal of the benefit of exploiting multiple mapping images into a feed-forward localization model.
>
> Regarding the accuracy of FastForward on larger-scale datasets, we kindly refer the reviewer to the reply to reviewer yDpi, where we address this topic.

---

> > ### Author Response · Authors · 2025-12-03
> > **Updates in the Post-Rebuttal Version**
> >
> > We thank Reviewer Ryg4 for their constructive feedback. We have uploaded a revised version of the paper incorporating the reviewer's suggestions. Specifically:
> > - We added the 25cm, 5° accuracy threshold to the ablation study in Appendix C.2. As suggested, we included this metric because it is used on the Map-free leaderboard.
> > - We included a discussion regarding scaling FastForward to large areas in Appendix C.1.

---

### Official Review · Reviewer_nKKy · 2025-10-31

**Soundness:** 2
**Presentation:** 3
**Contribution:** 3
**Rating:** 2
**Confidence:** 5

**Summary:**

The paper consider the problem of visual localization, i.e., the task of estimating the camera pose of a given query image wrt. some scene representation. State-of-the-art localization approaches are typically based on some form of 3D representation of the scene. Depending on the exact representation, building these 3D representations can take minutes to hours. This work thus investigates a scene representation that can be build in seconds instead: Given a query image, image retrieval is used to find the most similar database images representing the scene. The pose of the query image is then predicted wrt. the retrieved database images in a single forward pass. Experiments on standard benchmark datasets show that the proposed approach outperforms other methods that follow a similar paradigm (not building an expensive 3D scene representation) but can potentially perform significantly worse than methods based on a 3D scene representation (see results for Cambridge Landmarks).

**Strengths:**

The paper tackles an important practical problem and proposes a sound solution.

The method is evaluated on standard benchmark datasets and shows good performance compared to the baseline methods.

The paper is in general easy to read and it is easy to follow its argumentation.

**Weaknesses:**

My main concern are missing discussions of and comparisons with prior work:

1. Minor: [Kulhanek et al., ViewFormer: NeRF-free Neural Rendering from Few Images Using Transformers, ECCV 2022] proposed a transformer architecture that takes N images and their known poses as well as either an additional image or a camera pose as input and, in a single forward pass, either predicts the pose of the additional image or a rendered view from the additional camera pose. In the main paper, they focus on the latter case, i.e., novel view synthesis, while their supp. mat. presents localization results. While their method is not competitive with the proposed one, it seems to me that the paper still should be included in the related work section.

2. Relatively minor: [Torii et al., Are Large-Scale 3D Models Really Necessary for Accurate Visual Localization?, TPAMI 2021] (based on [Sattler et al., Are Large-Scale 3D Models Really Necessary for Accurate Visual Localization?, CVPR 2017]) asked a similar question to the one posed in the work under submission: Do we really need pre-computed 3D models for accurate visual localization? Their answer is that such models are not necessarily needed. Given the similarities in questions and answers, it seems to make sense to discuss this prior work.

3. Major: An alternative to estimating the pose of a query image by estimating relative poses wrt. each retrieved database image is to compute the pose of the query wrt. multiple database images (in form of semi-generalized relative pose [Zheng & Wu, Structure from motion using structure-less resection, ICCV 2015] or semi-generalized homography [Bhayani et al., Calibrated and Partially Calibrated Semi-Generalized Homographies, ICCV 2021] estimation). The most simplest to implement approach is to estimate the pose of the query image by first estimating the relative pose to one database image from 5 point correspondences (e.g., by estimating the essential matrix) and then using a single additional point correspondence with another database image to recover the scale of the translation [Zhang & Wu, ICCV 2015]. This E5+1 solver has been implemented for quite some time in the PoseLib library and it is quite simple to integrate into a localization pipeline (essentially: perform image retrieval to identify relevant database images, compute 2D-2D matches between the query and the top-retrieved images, estimate the pose by calling PoseLib's E5+1 solver wrapped in RANSAC (PoseLib has this readily implemented). While not as accurate as methods based on 3D representations, it works quite well, as shown in [Panek et al., A Guide to Structureless Visual Localization, arXiv:2504.17636] (the paper compares multiple existing approaches for localization based on representing the scene by a set of images rather than a 3D model, i.e., methods that offer fast representation-computation times). Here are results on multiple datasets obtained by this approach using RoMa feature matching after retrieving the top-20 most relevant database images using EigenPlaces descriptors for retrieval:

**Cambridge Landmarks**

Reporting median position errors [cm] / median orientation errors [deg] / accuracy at 10cm and 10deg / accuracy at 20cm and 20deg / accuracy at 25cm and 2deg:

| method | Great Court | King’s College | Old Hospital | Shop Facade | St Mary’s Church |
| --- | --- | --- |  --- | --- | --- |
| FastForward | 62 / 4 / 5.1 / 15.5 / - | 24 / 4 / 15.5 / 42.9 / - | 26 / 0.5 / 17.6 / 41.8 / - | 8 / 4 / 64.1 / 92.2 / - | 14 / 0.5 / 28.3 / 75.7 / - |
| E5+1 | 29 / 0.1 / - / - / 43.2 | 19 / 0.3 / - / - / 60.6 | 24 / 0.5/ - / - / 51.6 | 6 / 0.3 / - / - / 95.1 | 10 / 0.3 / - / - / 86.8|

**Indoor 6**

Reporting median position errors [cm] / median orientation errors [deg] / accuracy at 5cm and 5deg / accuracy at 10cm and 10deg / accuracy at 20cm and 20deg

Results for E5+1 are reported as the average over the scenes.

|method | average |
| --- | --- |
| FastForward | 4 / 0.7 / - / 91.1 / 98.1 |
| E5+1 |  1 / 0.15 / 94.9 / - / - |

As can be seen, this simple baseline performs similar to FastForward and does not require any specific learning. The main potential advantage that I can see for FastForward is that it might be faster at run-time due to requiring only a single forward pass. Then again, one can always extract only a limited number of features per image (e.g., I have good experience extracting 256 or 512 aliked features per image in the context of visual localization), which will make matching and pose estimation faster. As such, I believe that this would need to be shown through detailed experiments.

**Questions:**

What is the advantage of using FastForward over the simple E5+1 baseline? Please provide detailed experiments with the E5+1 baseline (using aliked features with the LightGlue matcher) by varying the number of extracted features and reporting the pose accuracy-run-time tradeoff and comparing it to FastForward.

---

> ### Author Response · Authors · 2025-11-20
>
> We thank the reviewer for their time and significant effort, even providing experimental results for a new baseline. We agree with the reviewer that this baseline is intriguing, and should be discussed and compared to which we will do further below and in a final version.
>
> However, what we do not agree with, and are frankly quite disappointed by, is that this missing comparison is the main reason for a clear reject rating.
>
> The E5+1 baseline has been widely overlooked in the relocalization literature. The associated solver has been proposed in the context of SfM reconstruction in an ICCV 2015 paper (Zheng & Wu) with 100 citations in 10 years. The solver has been part of the PoseLib library since 2022 but a public implementation of an E5+1 relocalization pipeline does not exist. None of our various competitors compares to it or even mentions it. The only account for E5+1 in a relocalization context is the unpublished arXiv paper from earlier this year (Panek et al.). This report states "Interestingly, [the E5+1 solver] has not been used as a baseline in other works on structureless localization".
>
> We agree that the E5+1 solver should receive more attention, and we are happy to help by discussing it in a final version. We do not agree that missing it initially is a major negligence from our side.
>
> Nonetheless, we are convinced that FastForward represents substantial progress in structure-less relocalization, even with the E5+1 baseline taken into account.
>
> (1) Panek et al. [arXiv 2025] ascertained when comparing learning-based regression to different variations of classical feature matching: "relative pose regression-based approaches perform worst, despite strong recent progress". In fact, they performed so poorly in that report, despite excessive running times, that they were excluded from the main experiments after some initial tests. In contrast, the reviewer concludes that FastForward "performs similar" to E5+1 in their tests, meaning a significant step forward for learning-based pose regression. We would like to note that the reviewer made several mistakes when pasting our accuracy into their review, reporting 10x our rotation error for multiple scenes on the Cambridge dataset.
>
> Next, following the question of the reviewer, we provide the results for different configurations of the suggested baseline (E5+1 with ALIKED and LightGlue matcher) and FastForward in the Cambridge dataset. We report the median translation errors [m], median rotation errors [deg], accuracy at 10cm and 10deg [%], accuracy at 25cm and 2deg [%], and query latency time [s]:
> |Cambridge Dataset|Trans. err. (m) $\downarrow$ | Rot. err. (°) $\downarrow$ | Acc. 10cm, 10° $\uparrow$ | Acc. at 25cm, 2° $\uparrow$ | Query latency (s) $\downarrow$
> |:---:|:---:|:---:|:---:|:---:|:---:|
> E5+1 - ALIKED w/ LG (kpts: 128)|0.25|0.42|32.5|60.7|2.18
> E5+1 - ALIKED w/ LG (kpts: 256)|0.22|0.36|35.6|63.7|2.35
> E5+1 - ALIKED w/ LG (kpts: 512)|0.20|0.34|37.4|65.6|2.45
> E5+1 - RoMa|0.18|0.30|–|67.5|19.0
> FF (Paper)|0.27|0.43|26.1|60.1|2.80
> FF (5k Pts)|0.27|0.42|26.7|60.8|0.49
>
> FastForward is the fastest method, localizing a new query image in only 0.49s. We got the latency down to 0.49s just by randomly subsampling the 2D-3D correspondences before PnP-RANSAC. In terms of accuracy, we agree that methods such as E5+1 with RoMa matcher perform better; however, the latency significantly increases (+3700%), while only improving its accuracy by 11%. E5+1 with ALIKED+LG offers a middle ground; it improves by 8% (512 kpts) and 5% (256 kpts) FastForward’s accuracy, but also takes 400% and 380%, respectively, longer to localize. Note that FastForward slightly outperforms ALIKED-LG with 128 kpts (25cm, 2deg) while providing faster localization. We believe that FastForward advances in structure-free localization in terms of both latency and competitive accuracy.
>
> We also report results in the Indoor6r dataset:
> |Indoor6 Dataset|Trans. err. (m) $\downarrow$ | Rot. err. (°) $\downarrow$ | Acc. 10cm, 10° $\uparrow$ | Acc. at 25cm, 2° $\uparrow$ | Query latency (s) $\downarrow$
> |:---:|:---:|:---:|:---:|:---:|:---:|
> | E5+1 - ALIKED w/ LG (kpts: 128) | 0.05 | 0.85 | 73.2 | 76.6 | 0.57 |
> | E5+1 - ALIKED w/ LG (kpts: 256) | 0.04 | 0.62 | 80.9 | 83.1 | 0.60 |
> | FastForward (Paper) | 0.04 | 0.72 | 91.1 | 93.1 | 2.50 |
> | FastForward (5k Pts) | 0.04 | 0.62 | 91.5 | 94.2 | 0.35 |
>
> In the Indoor6 dataset, FastForward outperforms ALIKED-LG, both in terms of accuracy and latency. As reported by the reviewer, E5+1 with RoMa obtains lower median errors but significantly higher localization time.

---

> > ### Author Response · Authors · 2025-11-20
> >
> > (2) On top of reasonable accuracy on classical benchmarks such as Cambridge or Indoor6, FastForward excels in scenarios that require extreme robustness, like front-to-back localization. We kindly ask the reviewer to consider the video "wayspots_lawn.mp4" in the supplement. We think this kind of robustness to extreme viewpoint changes is hard to achieve with approaches based on classical 2D-2D feature matching.
> >
> > See below the results in the Wayspots dataset:
> > |Wayspots Dataset|Trans. err. (m) $\downarrow$ | Rot. err. (°) $\downarrow$ | Acc. 10cm, 10° $\uparrow$ | Acc. at 25cm, 2° $\uparrow$ | Query latency (s) $\downarrow$
> > |:---:|:---:|:---:|:---:|:---:|:---:|
> > | E5+1 - ALIKED (kpts: 256) | 1.84 | 18.25 | 30.2 | 35.9 | 0.97 |
> > | E5+1 - ALIKED (kpts: 512) | 1.25 | 13.54 | 38.5 | 44.3 | 1.00 |
> > | E5+1 - ALIKED (kpts: 1024) | 0.65 | 8.13 | 45.3 | 51.7 | 1.09 |
> > | E5+1 - ALIKED (kpts: 2048) | 0.41 | 5.13 | 45.9 | 54.3 | 1.39 |
> > | E5+1 - RoMa | 0.77 | 4.12 | 49.5 | 58.9 | 19.45 |
> > | FastForward (Paper) | 0.17 | 1.79 | 51.0 | 61.1 | 2.80 |
> > | FastForward (5k Pts) | 0.17 | 1.75 | 51.4 | 63.0 | 0.54 |
> >
> > FastForward outperforms the E5+1 baseline even when it is paired with the state-of-the-art 2D-2D feature matcher RoMa. Besides, FastForward localizes a new query image in 0.54s, while E5+1 with RoMa takes almost 20 seconds. FastForward proves to be more accurate and faster than any of the proposed E5+1 solver baselines in these extreme scenes.
> >
> > (3) As the reviewer points out, we see a clear advantage of FastForward in terms of run time, since it avoids 21x evaluation of feature extraction and matching of the query to all retrieved mapping images. As discussed above, FastForward improves the latencies of methods based on the E5+1 solver by 3700% and 400% when paired with RoMa or ALIKED-LG, respectively.
> >
> > Lastly, we agree with the reviewer about the other missing references and will include them in the related work section.

---

> > > ### Comment · Reviewer_nKKy · 2025-11-20
> > >
> > > Thank you very much for this additional experiment.
> > >
> > > If I remember correctly, [Barroso-Laguna et al., Matching 2D Images in 3D: Metric Relative Pose from Metric Correspondences, ECCV 2024] show that 2D-2D correspondences can be established even under very strong viewpoint changes. Why not integrate these features into the E5+1 baseline? I feel that for the claim that "FastForward excels in scenarios that require extreme robustness, like front-to-back localization", such a comparison would be necessary to validate the claim.

---

> > > > ### Author Response · Authors · 2025-11-24
> > > > **Replying to Official Comment by Reviewer nKKy**
> > > >
> > > > We thank the reviewer for the prompt response and the detailed justification for their initial rating. We agree that baselines should not be selected based on popularity, and how a reviewer decides to weigh in the prominence of a baseline into their rating is up to them. We merely wanted to explain why we have overlooked this particular baseline, like many papers before us. We are happy to provide the results and discussions that support the benefits and drawbacks of using FastForward.
> > > >
> > > >
> > > > *” (1) What causes the large gap in query latency between E5+1 on Cambridge (2+ seconds) and on Indoor6 (around 0.6 seconds, much closer to FastForward).”*
> > > >
> > > > ​We believe the latency gap primarily stems from the differing number of retrieved images used in each benchmark. We use the top 20 retrieved images in the outdoor datasets (Cambridge) and the top 10 in the indoor datasets (Indoor6). See Sec. 4.3 from the main paper. Therefore, in the Indoor6 dataset, both FastForward and the E5+1 baselines receive 10 mapping images (rather than 20), which accelerates the query pose computation (e.g., by reducing the number of images E5+1 must match).
> > > >
> > > > *” (2) What are the technical parameters (query image resolution, number of retrieved images, number or RANSAC iterations) for E5+1?”*
> > > >
> > > > **Number of retrieved images**
> > > >
> > > > We integrate the E5+1 baseline into our localization pipeline with minimum changes, ensuring that common parameters, such as image retrieval settings, remain constant between FastForward and E5+1. Thus, as mentioned, we use the top 20 mapping images for Cambridge and the top 10 for Indoor6 for FastForward and the E5+1 baselines.
> > > >
> > > > **Query image resolution**
> > > >
> > > > For E5+1 with ALIKED-LG, we use the default configuration provided in LightGlue, which resizes the largest dimension to 1024. That is, if height > width, then the image will be rescaled to (size, size * width / height) with size=1024. In FastForward, we rescale the largest dimension to 512 and center-crop to the closest training resolution (e.g., (512, 384), (512, 336), (512, 288), (512, 256), or (512, 160)).
> > > >
> > > > **RANSAC**
> > > >
> > > > For E5+1, we use an epipolar error threshold of 12 px (as in [Panek et al., A Guide to Structureless Visual Localization]) and a maximum of 10,000 iterations. For FastForward's pose solver, we use the same hyperparameters proposed in DUSt3R. Specifically, we define the reprojection error threshold as reproj_err_diagonal_ratio * math.sqrt(W^2 + H^2), where reproj_err_diagonal_ratio = 0.008. For instance, in Wayspots, the reprojection error threshold is 5.12px. We also use a maximum of 10,000 iterations.
> > > >
> > > >
> > > > *” (3) Accuracy under the 5cm, 5deg threshold for Indoor6*
> > > >
> > > > As suggested by the reviewer, we have also computed the accuracy under the 5cm, 5° threshold in the Indoor6 dataset for completeness:
> > > > |Indoor6 Dataset | Acc. 5cm, 5° $\uparrow$ | Acc. 10cm, 10° $\uparrow$ | Acc. at 25cm, 2° $\uparrow$ | Query latency (s) $\downarrow$
> > > > |:---:|:---:|:---:|:---:|:---:|
> > > > | E5+1 - ALIKED w/ LG (kpts: 128) | 55.5 | 73.2 | 76.6 | 0.37 |
> > > > | E5+1 - ALIKED w/ LG (kpts: 256) | 64.4 | 80.9 | 83.1 | 0.40 |
> > > > | FastForward (Paper) | 69.3 | 91.1 | 93.1 | 2.50 |
> > > > | FastForward (5k Pts) | 69.6 | 91.5 | 94.2 | 0.35 |
> > > >
> > > >
> > > > As with the other metrics, the same trend is observed under the 5cm, 5° threshold: FastForward outperforms E5+1 with ALIKED-LG accuracy while being faster. E5+1 with RoMa provides the most accurate poses (94.9%), but is significantly slower (11.01s versus 0.35s).
> > > >
> > > >
> > > > *” (4) Comparison to MicKey [Barroso-Laguna et al., Matching 2D Images in 3D: Metric Relative Pose from Metric Correspondences]”*
> > > >
> > > > We provide the results using the E5+1 solver with MicKey keypoint matches:
> > > >
> > > > |Wayspots Dataset|Trans. err. (m) $\downarrow$ | Rot. err. (°) $\downarrow$ | Acc. 10cm, 10° $\uparrow$ | Acc. at 25cm, 2° $\uparrow$
> > > > |:---:|:---:|:---:|:---:|:---:|
> > > > | E5+1 w/ MicKey | 2.03 | 13.30 | 27.3 | 30.6 |
> > > > | FastForward| 0.17 | 1.75 | 51.4 | 63.0 |
> > > >
> > > > We would like to note that Wayspots is part of the Map-free training set. Thus, MicKey was trained on those scenes. Nevertheless, FastForward outperforms E5+1 MicKey in Wayspots. As stated by the reviewer, MicKey handles extreme viewpoint differences, but it exhibits lower performance under strict acceptance thresholds, as seen in the Map-free leaderboard. One explanation could be due to its training supervision and objective: MicKey was trained with only pose supervision and end-to-end optimized for the metric relative pose task, not as a 2D-2D matcher. MicKey proved effective when estimating metric pose from only a single map image. In contrast, FastForward was trained with camera poses and depth maps, a significantly stronger supervision signal than that used by MicKey. Moreover, FastForward was designed to exploit multiple mapping images in its prediction, which proved to be effective in the relocalization task.
> > > >
> > > >
> > > > Finally, we thank the reviewer for the references provided. They will be added to the related work section.

---

> > > > > ### Comment · Reviewer_nKKy · 2025-11-25
> > > > >
> > > > > Thank you very much for the clarifications and the additional experiments.
> > > > >
> > > > > It seems that the large differences (>2 seconds vs. around 0.5 seconds) were due to data loading as the time required for Cambridge has dropped.
> > > > >
> > > > > Looking at the results on Indoor-6 and Wayspots, it is clear that FastForward performs better in terms of pose accuracy. I don't think that executing times compared to E5+1 needs to be a selling point in this case (then again, you might want to adjust the name of the method). If you want to make a claim about faster processing times, I would feel more comfortable if you experiment with running feature extraction at lower resolutions (640 pixels should be fine; I have seen examples where going down to 320 pixels still gives good results) (which will make extraction times faster) and running fewer RANSAC iterations (e.g., 500, 1000, 5000). As things stand, the run-time difference is not too large. Looking at the timings for E5+1 (when varying the number of extracted features and the number of retrieved images), I would expect that the limiting factor in terms of times is RANSAC. Using fewer iterations can reduce run-times without negatively impacting pose accuracy. I would feel more comfortable with claiming superior run-time efficiency if the baseline is tuned a bit.
> > > > >
> > > > > If you show such an experiment, or if you drop the claim about a higher run-time efficiency, I'd be happy to champion the paper for acceptance.

---

> > > > > > ### Author Response · Authors · 2025-11-27
> > > > > >
> > > > > > We would like to thank the reviewer, nKKy, for working with us to improve the paper. We appreciate the constructive discussion.
> > > > > >
> > > > > > We have further tuned the E5+1 baseline with ALIKED-LightGlue matches. First of all, we report results when extracting 256 keypoints with ALIKED, which we believe offers a good trade-off between accuracy and latency. We vary the image size and the maximum number of RANSAC iterations:
> > > > > > | Cambridge Dataset | Trans. err. (m) $\downarrow$ | Rot. err. (°) $\downarrow$ | Acc. 10cm, 10° $\uparrow$ | Acc. at 25cm, 2° $\uparrow$ | Query latency (s) $\downarrow$ |
> > > > > > |:---:|:---:|:---:|:---:|:---:|:---:|
> > > > > > | FastForward (RANSAC: 10000) | 0.27 | 0.42 | 26.7 | 60.8 | 0.49 |
> > > > > > | ALIKED-LG - Im: 1024 - RANSAC: 10000 | 0.22 | 0.36 | 35.6 | 63.7 | 0.82 |
> > > > > > | ALIKED-LG - Im: 640 - RANSAC: 10000 | 0.21 | 0.33 | 35.9 | 66.0 | 0.70 |
> > > > > > | ALIKED-LG - Im: 640 - RANSAC: 5000 | 0.21 | 0.33 | 35.9 | 66.0 | 0.70 |
> > > > > > | ALIKED-LG - Im: 640 - RANSAC: 1000 | 0.21 | 0.33 | 35.9 | 65.9 | 0.70 |
> > > > > > | ALIKED-LG - Im: 640 - RANSAC: 500 | 0.21 | 0.33 | 35.9 | 66.0 | 0.63 |
> > > > > > | ALIKED-LG - Im: 640 - RANSAC: 100 | 0.21 | 0.34 | 36.0 | 65.8 | 0.56 |
> > > > > > | ALIKED-LG - Im: 320 - RANSAC: 5000 | 0.20 | 0.34 | 35.8 | 65.0 | 0.72 |
> > > > > > | ALIKED-LG - Im: 320 - RANSAC: 1000 | 0.20 | 0.34 | 35.8 | 65.0 | 0.72 |
> > > > > > | ALIKED-LG - Im: 320 - RANSAC: 500 | 0.20 | 0.34 | 35.8 | 64.8 | 0.65 |
> > > > > > | ALIKED-LG - Im: 320 - RANSAC: 100 | 0.20 | 0.34 | 35.7 | 64.2 | 0.58 |
> > > > > >
> > > > > > In the table, we presented results for three different image resolutions, i.e., 1024, 640, and 320 pixels. As suggested by the reviewer, we see a latency improvement when doing feature extraction at lower resolutions. We do not see a benefit of further downsampling the image, i.e., to 320 pixels or below. We can observe that the accuracy does not decrease, but neither does its query latency. Feature extraction is faster for 320px images than for 640px images; however, matching times increase. For instance, consider the extraction and matching times for ALIKED with LightGlue (256kpts and 5000 RANSAC iterations):
> > > > > >
> > > > > > 640px max image size:
> > > > > > - Feature extraction time = 0.19s
> > > > > > - Matching time = 0.33s
> > > > > > - Solver time = 0.18s
> > > > > >
> > > > > > 320px max image size:
> > > > > > - Feature extraction time = 0.17s
> > > > > > - Matching time = 0.36s
> > > > > > - Solver time = 0.19s
> > > > > >
> > > > > > Besides the image resolution, we also provide insights when varying the maximum number of RANSAC iterations. For instance, reducing the number of iterations to 500 or 100 shows faster latency times; however, none of the proposed baselines is faster than FastForward.
> > > > > >
> > > > > > Based on previous experiments, we present another set of results when extracting 128 keypoints with the ALIKED feature extractor:
> > > > > > | Cambridge Dataset | Trans. err. (m) $\downarrow$ | Rot. err. (°) $\downarrow$ | Acc. 10cm, 10° $\uparrow$ | Acc. at 25cm, 2° $\uparrow$ | Query latency (s) $\downarrow$ |
> > > > > > |:---:|:---:|:---:|:---:|:---:|:---:|
> > > > > > | FastForward (RANSAC: 10000) | 0.27 | 0.42 | 26.7 | 60.8 | 0.49 |
> > > > > > | ALIKED-LG - Im: 1024 - RANSAC: 10000 | 0.25 | 0.42 | 32.5 | 60.7 | 0.65 |
> > > > > > | ALIKED-LG - Im: 640 - RANSAC: 10000 | 0.23 | 0.39 | 35.0 | 62.9 | 0.61 |
> > > > > > | ALIKED-LG - Im: 640 - RANSAC: 5000 | 0.23 | 0.39 | 35.0 | 62.9 | 0.61 |
> > > > > > | ALIKED-LG - Im: 640 - RANSAC: 1000 | 0.23 | 0.39 | 35.0 | 62.8 | 0.61 |
> > > > > > | ALIKED-LG - Im: 640 - RANSAC: 500 | 0.23 | 0.39 | 34.9 | 62.7 | 0.59 |
> > > > > > | ALIKED-LG - Im: 640 - RANSAC: 100 | 0.23 | 0.39 | 34.7 | 62.4 | 0.54 |
> > > > > >
> > > > > > In this case, the baseline can further reduce the latency to 0.54s while still providing good results. Lastly, we acknowledge that more configurations could be explored to surpass FastForward latency, and, therefore, we compute the results when extracting 64 keypoints and 100 maximum RANSAC iterations (640 image size):
> > > > > >
> > > > > > | Cambridge Dataset | Trans. err. (m) $\downarrow$ | Rot. err. (°) $\downarrow$ | Acc. 10cm, 10° $\uparrow$ | Acc. at 25cm, 2° $\uparrow$ | Query latency (s) $\downarrow$ |
> > > > > > |:---:|:---:|:---:|:---:|:---:|:---:|
> > > > > > | FastForward (RANSAC: 10000) | 0.27 | 0.42 | 26.7 | 60.8 | 0.49 |
> > > > > > | ALIKED-LG - 64kpts | 0.31 | 0.47 | 26.8 | 55.7 | 0.54 |
> > > > > >
> > > > > > This very light configuration provides lower pose accuracy than FastForward while not surpassing its latency.
> > > > > >
> > > > > > FastForward is fast and accurate; it provides more accurate poses than Reloc3r across all datasets and it is better than E5+1 with ALIKED-LG on Indoor6 and Wayspots. However, we agree that the E5+1 baseline or other localizers, e.g., Reloc3r, provide comparable latencies to FastForward. Therefore, we will check the paper for any exaggerated claims regarding runtime and correct them as appropriate.

---

> > > > > > > ### Comment · Reviewer_nKKy · 2025-11-28
> > > > > > >
> > > > > > > Thank you very much for this detailed additional experiment and the fruitful discussion of this work.
> > > > > > >
> > > > > > > To me, that resolves my remaining concerns and I am happy to raise my score to an 8. Overall, with the additional experiments performed by the authors and the additional discussions of related work promised by the authors, I think this is a good paper that is very clearly above the acceptance threshold and will make a very good paper for ICLR.

---

> > > > > > > > ### Author Response · Authors · 2025-12-03
> > > > > > > > **Updates in the Post-Rebuttal Version**
> > > > > > > >
> > > > > > > > We thank Reviewer nKKy for their constructive feedback and the fruitful discussion. We have uploaded a revised version of the paper incorporating these updates:
> > > > > > > > - We added the E5+1 baseline results and discussions across all benchmarks.
> > > > > > > > - We included an ablation study of the ALIKED-LG matcher configurations with the E5+1 solver in Appendix C.6. This ablation investigates the impact of the number of keypoints and RANSAC iterations on accuracy and latency.
> > > > > > > > - We incorporated all suggested references into the Introduction and Related Work sections (e.g., see the new subsection “Semi-generalized Relative Pose Estimation” starting on line 141).

---

> > > ### Author Response · Authors · 2025-11-24
> > >
> > > We noticed an error in the latency calculation for the E5+1 baselines presented in our previous comment and sincerely apologize for this oversight. The initial reported timings included the image loading and preparation overhead, which is embedded within the compute matches function. In FastForward, the data preparation is done in the dataloader, which is not taken into account for the time computation. We have now excluded this data preparation time and corrected the previous tables. Even so, FastForward still provides the fastest localizations. For instance, in Cambridge, it is 3470% faster than E5+1 RoMa and 88% faster than ALIKED-LG (512). Furthermore, FastForward also proves more accurate than E5+1 ALIKED-LG in Indoor6 and all the E5+1 baselines in the Wayspots dataset.
> > >
> > > See below the corrected tables:
> > >
> > > **Cambridge dataset:**
> > > |Cambridge|Trans. err. (m) $\downarrow$ | Rot. err. (°) $\downarrow$ | Acc. 10cm, 10° $\uparrow$ | Acc. at 25cm, 2° $\uparrow$ | Query latency (s) $\downarrow$
> > > |:---:|:---:|:---:|:---:|:---:|:---:|
> > > E5+1 - ALIKED w/ LG (kpts: 128)|0.25|0.42|32.5|60.7|0.65
> > > E5+1 - ALIKED w/ LG (kpts: 256)|0.22|0.36|35.6|63.7|0.82
> > > E5+1 - ALIKED w/ LG (kpts: 512)|0.20|0.34|37.4|65.6|0.92
> > > E5+1 - RoMa|0.18|0.30|–|67.5|17.5
> > > FF (Paper)|0.27|0.43|26.1|60.1|2.80
> > > FF (5k Pts)|0.27|0.42|26.7|60.8|0.49
> > >
> > > **Indoor6 dataset:**
> > > |Indoor6|Trans. err. (m) $\downarrow$ | Rot. err. (°) $\downarrow$ | Acc. 10cm, 10° $\uparrow$ | Acc. at 25cm, 2° $\uparrow$ | Query latency (s) $\downarrow$
> > > |:---:|:---:|:---:|:---:|:---:|:---:|
> > > | E5+1 - ALIKED w/ LG (kpts: 128) | 0.05 | 0.85 | 73.2 | 76.6 | 0.37 |
> > > | E5+1 - ALIKED w/ LG (kpts: 256) | 0.04 | 0.62 | 80.9 | 83.1 | 0.40 |
> > > | FastForward (Paper) | 0.04 | 0.72 | 91.1 | 93.1 | 2.28 |
> > > | FastForward (5k Pts) | 0.04 | 0.62 | 91.5 | 94.2 | 0.35 |
> > >
> > > **Wayspots dataset:**
> > > |Wayspots|Trans. err. (m) $\downarrow$ | Rot. err. (°) $\downarrow$ | Acc. 10cm, 10° $\uparrow$ | Acc. at 25cm, 2° $\uparrow$ | Query latency (s) $\downarrow$
> > > |:---:|:---:|:---:|:---:|:---:|:---:|
> > > | E5+1 - ALIKED (kpts: 256) | 1.84 | 18.25 | 30.2 | 35.9 | 0.69 |
> > > | E5+1 - ALIKED (kpts: 512) | 1.25 | 13.54 | 38.5 | 44.3 | 0.71 |
> > > | E5+1 - ALIKED (kpts: 1024) | 0.65 | 8.13 | 45.3 | 51.7 | 0.80 |
> > > | E5+1 - ALIKED (kpts: 2048) | 0.41 | 5.13 | 45.9 | 54.3 | 1.10 |
> > > | E5+1 - RoMa | 0.77 | 4.12 | 49.5 | 58.9 | 17.95 |
> > > | FastForward (Paper) | 0.17 | 1.79 | 51.0 | 61.1 | 2.80 |
> > > | FastForward (5k Pts) | 0.17 | 1.75 | 51.4 | 63.0 | 0.54 |

---

> > ### Comment · Reviewer_nKKy · 2025-11-20
> >
> > I very much thank the authors for the detailed reply and the additional experiments.
> >
> > I am sorry for the errors with the orientation errors for the Cambridge Landmarks dataset. Best guess is that when I converted from position errors in m to errors in cm, I accidentally also deleted the leading zeros for the orientation errors.
> >
> > I understand that the authors are disappointed by my initial rating and I will try to explain it better: My initial rating is based on the observation that a simple, rather old baseline, outperforms the proposed approach, without any specific training for the task at hand, in terms of pose accuracy. Naturally, this raises the question what the advantage of the proposed approach is? I.e., what is the contribution of the work to the literature? I felt that without comparing against this baseline in detail, these important questions remain unanswered. Hence, I felt that the paper, in the form that it was submitted was not ready for publication as it was missing crucial parts.
> >
> > The fact that the baseline was not included by competitors is an explanation why the baseline was not included, but it is not a reasonable justification that it should not be included (I am sure the authors do not want to suggest that baselines should be chosen based on their popularity, which is a malicious way of reading the reply ;) ). Set aside that the arXiv paper in question has been available about 5 months before the ICLR submission deadline, it is not the "only account for E5+1 in a relocalization context":
> > * It was used as a baseline in [Bhayani et al., Calibrated and Partially Calibrated Semi-Generalized Homographies, ICCV 2021].
> > * Both the papers by Zheng & Wu and by Bhayani et al. are mentioned in the ICCV 2021 tutorial "Large-Scale Visual Localization"  by Brachmann et al. (see about 2:34:47 to 2:37:17 here: https://www.youtube.com/watch?v=RaVPiIGhdWk).
> > * The E5+1 baseline was explicitly used in [Panek et al., Combining Absolute and Semi-Generalized Relative Poses for Visual Localization, arXiv:2409.14269] (published at DAGM GCPR 2025]), which has been available about 1 year before the ICLR 2026 submission deadline.
> > * The E5+1 baseline is one particular example for semi-generalized relative pose estimation. Semi-generalized relative pose estimation is the problem of estimating the pose of a central camera (i.e., a camera where all viewing rays meet in a single point) relative to a generalized camera (i.e., a camera where not all viewing rays meet in a single point). Essentially, it is the problem of estimating the relative pose from correspondences between 2D pixel positions and 3D rays. This is the same pose estimation setting that arises in the context of privacy-preserving visual localization, when replacing 3D point clouds with 3D line clouds [Speciale et al., Privacy Preserving Image-Based Localization, CVPR 2019].
> > * Semi-generalized relative pose estimation is a special case of generalized relative pose estimation (estimating the relative pose between two generalized cameras), which is a well-studied problem in classical 3D computer vision.
> >
> > In short, there have been enough opportunities through which the authors could have learned about this baseline. In any case, a baseline not being widely used in a specific line of work on visual localization does not invalidate the baseline nor does it mean that failure to compare against it shouldn't be grounds for rejection if there is no clear evidence that a proposed approach outperforms this baseline.
> >
> > That being said, the new experiments provided by the authors address my main concern, i.e., showing an advantage over the E5+1 baseline. Looking at the results, it is clear that the advantage is in run-time, which potentially comes at the cost of pose accuracy (at least this is the case for the finer 10cm, 10deg threshold for Cambridge Landmarks; it is a pity that the authors did not include the commonly used 5cm, 5deg threshold for Indoor6).
> >
> > Based on these new experiments, I am happy to increase my rating to an accept rating. However, there are several things that are not fully clear to me and that I would like to know to better understand the results:
> > * What causes the large gap in query latency between E5+1 on Cambridge (2+ seconds) and on Indoor6 (around 0.6 seconds, much closer to FastForward).
> > * What are the technical parameters (query image resolution, number of retrieved images, number or RANSAC iterations) for E5+1?

---

### Author Response · Authors · 2025-12-03
**Summary of the discussion (before it was locked)**

We thank the reviewers and ACs for their time and effort. Below, we provide a summary of the discussion.

Initially, we received borderline reviews with two accept ratings (Ryg4: 8, yDpi: 6) and two reject ratings (nKKy: 2, 13TE: 2). After an extensive discussion with reviewer nKKy about comparing to one more baseline, the reviewer changed their assessment from reject to accept (from 2 to 8). Referring to the same discussion, reviewer yDpi confirmed their positive score with a tendency to raise it further.

The only remaining reject rating by reviewer 13TE was given with a generic "lack of novelty" reasoning. Unfortunately, the reviewer did not respond to our rebuttal in time. However, all other reviewers rated the contribution of the draft as "good".

We incorporated the reviewer feedback into an updated version of the draft. A summary of changes can be found in each reviewer's thread.

---

### Meta-Review · Area_Chair_8Qcv · 2025-12-12

**Summary:**

The paper received 8,6,2,2 initially.

The authors provided responses to the expressed concerns.

Reviewer nKKy was satisfied with the rebuttal and improved from 2 to 8.

Reviewer yDpi confirmed the 6 rating with tendency to further raise it.

Reviewers Ryg4 (rating 8) and 13TE (rating 2) did not participate in discusses, had no post-rebuttal updated recommendation.

After carefully checking the paper, the reviews and the responses from the authors, the ACs agree with the majority of the reviewers that the paper makes significant contributions and the authors are invited to integrate contents from their responses into the camera ready paper.

**Reviewer Concerns:**

The paper received 8,6,2,2 initially.

The authors provided responses to the expressed concerns.

Reviewer nKKy was satisfied with the rebuttal and improved from 2 to 8.

Reviewer yDpi confirmed the 6 rating with tendency to further raise it.

Reviewers Ryg4 (rating 8) and 13TE (rating 2) did not participate in discusses, had no post-rebuttal updated recommendation.

After carefully checking the paper, the reviews and the responses from the authors, the ACs agree with the majority of the reviewers that the paper makes significant contributions and the authors are invited to integrate contents from their responses into the camera ready paper.

**Reviewer Scores:**

The paper received 8,6,2,2 initially.

The authors provided responses to the expressed concerns.

Reviewer nKKy was satisfied with the rebuttal and improved from 2 to 8.

Reviewer yDpi confirmed the 6 rating with tendency to further raise it.

Reviewers Ryg4 (rating 8) and 13TE (rating 2) did not participate in discusses, had no post-rebuttal updated recommendation.

After carefully checking the paper, the reviews and the responses from the authors, the ACs agree with the majority of the reviewers that the paper makes significant contributions and the authors are invited to integrate contents from their responses into the camera ready paper.

---

### Decision · Program_Chairs · 2026-01-26

Accept (Poster)